# *Artemisia absinthium* L.—Importance in the History of Medicine, the Latest Advances in Phytochemistry and Therapeutical, Cosmetological and Culinary Uses

**DOI:** 10.3390/plants9091063

**Published:** 2020-08-19

**Authors:** Agnieszka Szopa, Joanna Pajor, Paweł Klin, Agnieszka Rzepiela, Hosam O. Elansary, Fahed A. Al-Mana, Mohamed A. Mattar, Halina Ekiert

**Affiliations:** 1Chair and Department of Pharmaceutical Botany, Medical College, Jagiellonian University, Medyczna 9, 30-688 Kraków, Poland; asiek.pajor@student.uj.edu.pl; 2Family Medicine Clinic, Medizinisches Versorgungszentrum (MVZ) Burgbernheim GmbH, Gruene Baumgasse 2, 91593 Burgbernheim, Germany; bag-burgbernheim@gmx.de; 3Museum of Pharmacy, Medical College, Jagiellonian University, Floriańska 25, 31-019 Kraków, Poland; agnieszka.rzepiela@uj.edu.pl; 4Plant Production Department, College of Food and Agriculture Sciences, King Saud University, Riyadh 11451, Saudi Arabia; helansary@ksu.edu.sa (H.O.E.); falmana@ksu.edu.sa (F.A.A.-M.); 5Floriculture, Ornamental Horticulture, and Garden Design Department, Faculty of Agriculture (El-Shatby), Alexandria University, Alexandria 21545, Egypt; 6Department of Geography, Environmental Management, and Energy Studies, University of Johannesburg, APK Campus, Johannesburg 2006, South Africa; 7Department of Agricultural Engineering, College of Food and Agriculture Sciences, King Saud University, Riyadh 11451, Saudi Arabia; mmattar@ksu.edu.sa

**Keywords:** wormwood, chemical composition, biological activity, traditional applications, modern applications, safety of use

## Abstract

*Artemisia absinthium*—wormwood (Asteraceae)—is a very important species in the history of medicine, formerly described in medieval Europe as “*the most important master against all exhaustions*”. It is a species known as a medicinal plant in Europe and also in West Asia and North America. The raw material obtained from this species is *Absinthii herba* and *Artemisiae absinthii aetheroleum.* The main substances responsible for the biological activity of the herb are: the essential oil, bitter sesquiterpenoid lactones, flavonoids, other bitterness-imparting compounds, azulenes, phenolic acids, tannins and lignans. In the official European medicine, the species is used in both allopathy and homeopathy. In the traditional Asian and European medicine, it has been used as an effective agent in gastrointestinal ailments and also in the treatment of helminthiasis, anaemia, insomnia, bladder diseases, difficult-to-heal wounds, and fever. Today, numerous other directions of biological activity of the components of this species have been demonstrated and confirmed by scientific research, such as antiprotozoal, antibacterial, antifungal, anti-ulcer, hepatoprotective, anti-inflammatory, immunomodulatory, cytotoxic, analgesic, neuroprotective, anti-depressant, procognitive, neurotrophic, and cell membrane stabilizing and antioxidant activities. *A. absinthium* is also making a successful career as a cosmetic plant. In addition, the importance of this species as a spice plant and valuable additive in the alcohol industry (famous absinthe and vermouth-type wines) has not decreased. The species has also become an object of biotechnological research.

## 1. Introduction

Over the past few years, there has been an increase in interest in research on the chemistry and biological activities of *Artemisia* species. This is undoubtedly connected with the awarding of the Nobel Prize in Medicine in 2015 for the discovery of artemisinin―a sesquiterpenoid lactone effective in the treatment of malaria, found in *Artemisia annua* (annual mugwort). A commonly known species of the genus *Artemisia*, with an important place in the history of medicine, is *Artemisia absinthium* L.

The aim of this work was the review of the latest literature reports on *A. absinthium* in order to evaluate the importance of this species in the traditional phytotherapy, as well as in the modern medicine. The emphasis was put on the latest biological activities confirmed by scientific studies. The applications in the cosmetology and in the food industry were also announced. Moreover, the studies on the safety of use as well as the biotechnological researches were reported.

The information about *A. absinthium* was collected collected from various sources such as official websites e.g., The Plant List, GBIF (Global Biodiversity Information Facility), WHO (World Health Organization), FDA (Food and Drug Administration), EFSA (European Food Safety Authority), EMA (European Medicines Agency), CosIng (Cosmetic Ingredient database), classical books, databases of scientific journals (e.g., Scopus, PubMed, Google Scholar), on-line books, and pharmacopoeias. In addition, professional historical descriptions of this species were also analyzed.

*A. absinthium* has its natural habitats in Europe, West Asia, and North Africa. The species has been used for centuries as effective in various gastrointestinal ailments and in the treatment of helminthiases. Contemporary pharmacological studies have focused on confirming and determining the mechanisms of these traditional directions of activity. They have also demonstrated new, previously unknown possible therapeutic applications resulting from proven antiprotozoal, antibacterial, antifungal, anti-ulcer, hepatoprotective, anti-inflammatory, immunomodulatory, cytotoxic, analgesic, neuroprotective, anti-depressant, procognitive, neurotrophic, cell membrane stabilizing, and antioxidant effects.

Furthermore, *A. absinthium* has today an important place in the production of cosmetics. It also has an established position in the food industry, as a base for alcoholic beverages and as a spice. It has also become an object of biotechnological research.

Recent years have seen publication of review articles on this species. However, they present the existing, broad knowledge on its therapeutic values in a very general way [1,2].

While compiling this review, every effort was made to present in detail the qualities of this species, with particular emphasis on the current work on the chemistry of the plant, on the chemistry of its essential oil, its variability in chemical composition, mechanisms of action in the traditional applications and new directions of biological activities confirmed by scientific research, as well as current views on the safety of using this plant species. Additionally, the position of the plant in food industry and cosmetic industry was underlined and biotechnology investigations were presented.

## 2. General Information on the Species

*Artemisia absinthium* L.—wormwood (Asteraceae), is an herbaceous plant. This species has numerous (about 20) synonymous Latin names. The most common synonyms are: *Absinthium majus* Garsault [3,4,5] and *Absinthium officinale* Brot. [3,4,5], *Absinthium officinale* Lam. [6], *Absinthium vulgare* (L.) Lam. [3,4] *Absinthium vulgare* Gaertn. [6], *Artemisia absinthia* St.-Lag. [3,4]. Among the English names, the most popular is “wormwood”. It comes from the German word “wermet” meaning “keeping a clear mind” [7]. Some other English and foreign names are: absinth, absinth wormwood, absinthe, absinthium, Maderwood (English, USA), Absinth, Bitterer Beifuss, Wermkraut, Wermut, Wermutkraut, Wurmkraut (German), absinthe, grande absinthe, Herbe d’absinthe (French), Majri, Mastiyarah (Hindi), Yang ai, zhong ya ku hao (Chinese) [1,2,6,8,9,10,11]. *A. absinthium* is a shrub-like perennial plant growing to a height of 80 cm. In some habitats, it even reaches a height of up to 1.5 m. The whole plant is strongly pubescent and has an intense, sharp smell [1,7,12]. *A. absinthium* leaves have essential oil secreting hairs/glandular trichomes and covering T-hairs that have a protective function—they protect the plant against high temperatures and prolonged drought [7,13].

The stem is grey-green, strongly pubescent and ribbed—it usually has 5 flattened, longitudinal furrows [1,7,12]. The part of the stem with flowers reaches a diameter of no more than 2.5 mm [12].

The leaves also take on a grey-green color and are densely pubescent on both sides [1,7]. Their shape depends on where they are situated on the plant. The basal leaves have long petioles, and their blade is triangular or oval, bi- or tripinnatisect, the lower leaves are not as intensely divided, and the top leaves are lanceolate [12].

The capitulum inflorescences are gathered in loose panicles growing from the axils of the leaves. In these semicircular or circular heterogamous capitulum there are light-yellow ligulate female flowers, and tubular hermaphroditic flowers. The involucral bracts covering the capitulum are long and grey, with ensiform outer and oval inner leaves [1,12] The flowering period of the plant in Central Europe begins towards the end of July and lasts until October [7,12].

The fruit is a small achene with brown stripes.

*A. absinthium* comes from Europe, West Asia and North Africa. It is a species commonly found in Poland, Scotland and England. The species was introduced and acclimatized in North America and South America; it can also be found in Australia. In Kashmir, populations of *A. absinthium* occupy sites at an altitude of up to 2100 m. It occurs on roadsides, forest felling sites and clearings, as well as wasteland and stony ground [1,6,7,10,14].

*A. absinthium* reproduces mainly vegetatively by roots [7]. This species is not susceptible to pathogens, but the roots are sensitive to excessive irrigation, which quickly leads to rotting [7].

*A. absinthium* is a species cultivated today in the countries of southern Europe, the USA and Brazil.

The harvesting period begins with the appearance of the first flowers. Leafy shoots and basal leaves are cut off, while the woody parts are left behind. Harvesting can be done several times a year.

The drying process has a significant impact on the quality of *A. absinthium* essential oil. It has been observed that even slight heating of the air affects the organoleptic characteristics of the oil. In addition, the collected herb should not be spread in a thick layer because it then dries very slowly. Drying should be carried out in shaded airy rooms or drying chambers at room temperature [15].

## 3. Phytochemical Characteristics

*A. absinthium* contains numerous compounds responsible for its biological activities. The herb of this species is considered to be the raw material for oil extraction. The essential oil content of the herb varies both qualitatively and quantitatively depending on the geographical region and environmental conditions (Table 1). The concentration of oil in the plant ranges from 0.2% in a dry climate to 1.5% in a humid climate [16]. The highest concentration of essential oil in *A. absinthium* herb is observed in June and July [15]. To indicate the main component of the oil is difficult because the results of phytochemical tests are not conclusive. The most frequently listed compounds are thujyl alcohol esters, *α*-thujone, *β*-thujone, camphene, *α-*cadinene, guaiazulene (*Z*)-epoxyocimene, (*E*)-sabinyl acetate, (*Z*)-chrysantenyl acetate [1,16] (Figure 1). It has been noted that among populations growing in areas above 1000 m a.s.l. *α*-thujone is the characteristic compound, while (*Z*)-epoxyocimene dominates below this height [16].

The concentration and composition of *A. absinthium* essential oil from plants growing in Poland were tested in 2007. In the flowering herb of plants collected from an area in the Mrągowo District (North-East part of Poland), the essential oil content ranged from 0.90% to 1.45%. The concentration of the oil in plants in the vegetative stage was lower and amounted to 0.60–0.94%. The compounds isolated in the largest quantities were: sabinyl acetate, chrysantenyl acetate, *β*-thujone, and cineol [17].

Other important compounds of the herb of *A. absinthium* are bitter sesquiterpenoid lactones [15], of which the main metabolite is a guaianolide dimer—absinthin (0.2–0.28%) (Table 2). Other compounds are found in high concentration such as absinthin isomers—anabsinthin, anabsin, artabsin (0.04–0.16%), and absintholide (Figure 2) [16]. The highest concentration of bitter components is obtained from plants harvested in September [15]. Other bitter compounds that have been isolated from the plant include artamaridinin, artamarin, artamarinin and artamaridin [21].

*A. absinthium* extracts contain high concentrations of blue chamazulene [18,21,22,23,26,28]—a compound from the group of azulenes, resulting from the transformation of sesquiterpenoid matrix [33]. Other azulenes isolated from the herb are 3,6-dihydrochamazulene, 7-ethyl-1,4-dimethylazulene [19], 7-ethyl-5,6-dihydro-1,4-dimethylazulene, dihydrochamazulene isomer [16], prochamazulenogen [21], and azulene [1,21].

The plant also contains numerous flavonoids, including: quercetin, kaempferol, apigenin, artemethin, and rutoside [1,16,21,22,34,35], and numerous phenolic acids such as: chlorogenic, ferulic, gallic, caffeic, syringic, and vanillic, and derivatives of caffeoylquinic acid [16,21,36,37].

Other compounds found in smaller amounts are the chalcone—cardamonin [38,39], coumarins (herniarin, coumarin) [22,27], fatty acids [1], tannins [16,21,31], carotenoids, lignans [16,21] and resinous substances [31].

Four new, previously unknown structures have been isolated from *A. absinthium* herb by Javed et al.; they were glycosidic esters: 3,11-dimethyldodecan-1,7-dioic acid-1-*β*-D-glucopyranosyl-6′-octadec-9′′-enoate, lanost-24-en-3*β*-ol-11-one-28-oic acid-21,23 *α*-olide-3*β*-D-glucopyranosyl-2′-dihydrocaffeoate-6′-decanoate, stigmast-5,22-dien-3*β* -ol-21-oic acid-3*β*-glucopyranosyl-2′-octadec-9′′-enoate, and tricosan-14-on-1,4-olide-5-eicos-9′-enoate. The authors classified the first two compounds as sterols, while the other two compounds could not be classified into a specific group of metabolites [40].

The composition of *A. absinthium* extract depends on the extractant used. It has been proved that an ethanolic extract has a significantly higher concentration of flavonoids, phenols and tannins in comparison with aqueous and chloroform extracts [31].

## 4. Importance of *A. absinthium* in the History of Medicine

Wormwood has always been associated with a very bitter taste. In the Polish language, it is the proverbial quintessence of bitterness (in the popular saying “bitter as wormwood”), but already in ancient Greece it owed its name—*α*ψίνθιον, Apsinthion (Latin Apsinthium)—to its not very pleasant taste. Dioscorides (1st century AD) and Theophrastus (4th/3rd century BC) associated it with the Greek words “ápsinthos”—unpleasant, disagreeable, or “ápinthos”—unfit for drinking. In the Germanic literature on herbal medicine, the name “Wermut” appears, indicating the antiparasitic effect of this herb attributed to its bitter taste (“Werm” in Old German means “worm”) [44]. In Dioscorides’s *De materia medica*, wormwood, usually taken in the form of tincture, is described as having warming, astringent and stimulating effects, being able to relieve stomach and abdominal pains, and effective against poison [45]. Pliny the Elder (1st century AD) also recommends *Absinthium* as a hypnotic, laxative, menstruation-inducing agent, healing “fistulas on the eyes”, and even as a cosmetic—the ash from it combined with a rose ointment “blackens the hair” [46]. The therapeutic spectrum of wormwood (*Artemisia absinthium*) is very similar to that of mugwort (*Artemisia vulgaris*), together with which it has often been described [47]; compared to mugwort, wormwood has a slightly stronger effect on the digestive system [48]. Apart from the characteristic bitterness, another special property of wormwood has been known since antiquity—depending on the dose, it stimulates the central nervous system, causing even epileptic seizures and hallucinations [49]. Both Walafrid Strabo in his *“Hortulus”* (9th centuryAD) and Hildegard von Bingen in *“Physica”* three centuries later emphasize the restorative effect of wormwood [50]. Hildegard, the famous mystic, saint and Doctor of the Church, praises wormwood as “*the most important master against all exhaustions*” [51]. The authors of the Renaissance era, describing the therapeutic indications for wormwood, put the main emphasis on ailments of the digestive system. Adam Lonitzer (1551) recommends it for “*strengthening the stomach*” and “*improving appetite*”, and Leonard Fuchs (1543) for “*removing stool, winds, and pain in the gut*”. Nota bene, Fuchs’s work begins with a description of vermouth. Both authors introduce eye compresses, which are to remove eye hyperaemia and improve eyesight [52,53]. The Polish herbals by Stefan Falimirz [54] and Marcin from Urzędów mention the detoxifying and restorative, as well as curative, effects of wormwood in gastrointestinal, liver, and biliary tract diseases; it was also supposed to cure skin, ear and eye diseases, remove odour from the mouth, and “drive out worms” [55]. A similar application is given in the famous *Herbiarz* (herbal) by Szymon Syreniusz (1613) [56]. In turn, the great work of Tabernaemontanus, *Neuw Kreuterbuch* (1588), in which the description of herbs also begins with vermouth, as part of the development of a holistic way of thinking and treatment in medicine, sees the benefit of the choleretic action of wormwood, both in the treatment of jaundice and “*in angry women who are full of bile*” [57]. In the declining years of that philosophy, consistently supplanted by the emerging modern medicine, Joachim Zedler in his *Universal-Lexicon* (1732) praises wormwood mouthwash solutions as able to improve the smell of breath and to strengthen the gums and teeth [58]. This indicates a tendency to rediscover hygiene and to use those properties of wormwood that in evidence-based medicine (EBM) are termed “antiseptic” [59]. Based on the essential oil of wormwood, a very popular alcoholic drink—absinthe—was created in the 19^th^ century, an intoxicant whose excessive consumption leads to irreversible damage to the central nervous system. Because of this effect, modern medicine prohibits the consumption of the ethereic oil obtained from this plant, or any spirit based on wormwood, but sees the benefits of its action on the digestive system in the form of a herb [60]. The folk use of wormwood also deserves a mention: added to washing, its smell is supposed to repel moths, lice, and bed bugs. The same intense smell, also released during the burning of this plant, was the basis of its ritual use in funeral processions and in repelling evil spirits [61]. In folk medicine, wormwood was used for fever, having been poured over with aqua vitae (spirit), or soaked with pepper in vodka, wine or water, and for stomach pain [62], while in folk veterinary medicine, cooked wormwood was given to cattle to improve their appetite [63].

## 5. Application in Traditional Medicine

*A. absinthium* is one of the most recognizable species of the genus *Artemisia* in the world. In European folk medicine, this plant has been used for millennia for many different diseases, in particular for parasitic diseases and digestive ailments, and when fever occurred [7]. According to the recommended traditional use, the leaves are used to lower the temperature, and the flowers help in diseases of the stomach and helminthiases. *A. absinthium* tincture is valued as a tonic and digestive aid [1]. The wormwood herb has been used to treat jaundice, constipation, obesity, splenomegaly, and also to treat anaemia, insomnia and bladder diseases. It has also served as a remedy for injuries and non-healing wounds [1]. The plant has been used as a base for preparing ointments and balms for use on the skin [7]. The pharmacological activity of this species in indications such as anaemia, menstrual cramps, treatment of skin lesions and difficult-to-heal wounds has not been scientifically confirmed [9].

In Indian Unani medicine, *A. absinthium* is the main ingredient in the drug “Afsanteen”, which is used for chronic fever, hepatitis and oedema [1], while the traditional Chinese medicine (TCM) uses *A. absinthium* in cancer therapy as a means of reducing angiogenesis [64].

## 6. Position in Modern Allopathy and Homeopathy

The pharmacopoeial raw material is the dried *A. absinthium* herb (*Absinthii herba*). According to the latest editions (9th and 10th) of the European Pharmacopoeia and the 11th Polish Pharmacopoeia, it is recommended to harvest the herb from young plants—in their first year of vegetation, basal leaves are cut off, and from older plants—sparsely leaved, flowering shoot tips. One can also use a mixture of them, as well as whole or broken-up fragments of the *A. absinthium* plant. The raw material is standardized for essential oil content; in the dried herb, this content must not be less than 2 mL/kg [12]. In addition, the bitterness index of the raw material must not be less than 10,000 [12].

The herb of *A. absinthium* is used today mainly to improve digestion. The plant is part of the pharmacopoeial (11th PF) Polish national herbal mixture *Species digestivae* together with *Cichorii radix, Angelicae archangelicae radix, Carvi fructus* and *Gentianae radix*. The tincture of *A. absinthium* (*Absinthii tinctura*) also has its pharmacopoeial monograph [65].

The raw material can also be found in other herbal mixtures recommended for hypoacidity, digestive disorders, or lack of appetite. The remedy is mainly in the form of infusions recommended for use before or during eating.

Since 1984, *Absinthii herba* has had a monograph in the German Pharmacopoeia. According to the German guidelines, the raw material was recommended for loss of appetite, digestive problems, and bile secretion disorders. It was recommended to use an infusion prepared from 1–1.5 g of dried herb [11,66,67]. The daily dose should not exceed 2–3 g [11]. In addition, the German Pharmacopoeia also mentioned a tincture from the herb [68]. However, *Absinthii tinctura* is not mentioned in the latest (10th) edition of the European Pharmacopoeia [69].

According to ESCOP (European Scientific Cooperative on Phytotherapy), *Absinthii herba* may be used in digestive disorders. Anorexia is also mentioned among the indications. ESCOP recommends not to use the herb for a period longer than 3–4 weeks [30].

*Absinthii herba* has its monograph published by the European Medicines Agency (EMA), in which, on the basis of well-established use, it is recommended to use the raw material in temporary loss of appetite, mild dyspepsia, and in gastrointestinal disorders. The listed forms in which the raw material can be used include finely divided or powdered herbal substance, fresh juice or tincture from the herb. Commercial herbal preparations are made in solid or liquid forms, and finely divided herb is used in herbal teas. All these forms are for oral use only. Contrary to the ESCOP recommendations, the EMA specifies the maximum duration of use as 2 weeks [11].

The fresh, flowering herb of *A. absinthium* is classified in the European Pharmacopoeia and the French Pharmacopoeia as a homeopathic raw material. The tincture produced should contain a minimum of 0.05% *w*/*w* derivatives of hydroxycinnamic acid, expressed in terms of chlorogenic acid [70]. In homeopathy, the plant is recommended for hallucinations, nightmares, nervousness, insomnia, dizziness, and epileptic seizures [71].

*A. absinthium* activity profiles documented in scientific papers, and the mechanisms of action of the raw material, are presented below in the text and in Table 3.

## 7. Biological Activities Confirmed by Scientific Research

### 7.1. Long-Known Possible Applications Confirmed by Modern Scientific Research

#### 7.1.1. Effect of Stimulating Digestion

The bitterness-imparting compounds contained in wormwood herb enhance digestion through various mechanisms. The first of these is irritation of the nerve endings on the tongue, which causes reflex secretion of gastric juice. Another mechanism involves the stimulation of secretory nerves in the liver and pancreas, as a result of which increased production of bile and pancreatic juice can be observed. The bitter substances also have a direct effect on the stomach, leading to increased gastric movements and enlargement of small vessels of the mucosa. It has been proved that the compounds contained in the essential oil also increase the production of digestive juices and improve blood flow.

It is worth noting that the active compounds contained in *Absinthii herba* do not cause a significant improvement in secretion in healthy people. The pharmacological effect is mainly observed among patients with reduced digestive juice production.

A study attempted to confirm the mechanism responsible for the increased secretion of digestive juices after the application of preparations containing *A. absinthium*. The experiment was conducted with the participation of volunteers who received encapsulated cellulose water (control group) or an ethanolic extract of *A. absinthium* herb (study group). The participants had their cardiovascular parameters tested before and after ingestion. It was found that in the control group the strength of heart contraction and arterial pressure increased, while among the study group given 1500 mg of wormwood tincture, a decrease in peripheral vascular resistance and reduced cardiac output were observed. In the control group, induction of gastric digestion occurred by increasing the strength of heart contraction. By comparison, the consumption of bitter compounds induced the gastric phase by increasing peripheral vascular resistance associated with the activation of the sympathetic nervous system. A change in postprandial haemodynamics in the gastric phase of digestion along with increased blood supply was reported as a likely mechanism for stimulating digestion [72].

#### 7.1.2. Anthelmintic Effect

*A. absinthium* has been proven to exert an anthelmintic effect against organisms such as: *Trichinella spiralis*, *Ascaris suum*, *Trichostrongylus colubriformis*, *Haemonchus contortus*, as confirmed by the scientific research outlined below.

One of the experiments examined the effects of *A. absinthium* on *Trichinella spiralis* (trichina worm). Methanolic extracts from the herb of the plant were administered to rats suffering from trichinosis. After sacrificing the rodents, samples of their muscles were subjected to trichinoscopic examination and artificial digestion. Both methods determined the number of larvae in muscle tissue. Using 300 g/kg of *A. absinthium* extract, the percentage of larvae during the intestinal phase of the disease decreased by 63.5% in the tongue, by 37.7% in the diaphragm, by 46.2% in the quadriceps muscles, and by 60.5% in the biceps and triceps [76].

The same parasite was the object of further research. In the first of the experiments, *A. absinthium* essential oil was applied to a suspension containing *T. spiralis* larvae, and after 24 h their infectivity was checked by giving them to mice. Seven days after infection, the rodents were sacrificed and the number of adult worms in the intestinal mucosa was counted. Using the essential oil at 1 mg/mL, up to 99.99% larvicidal effectiveness was achieved.

In another experiment, a group of mice were orally infected with *T. spiralis* larvae. A day later, the animals were given *A. absinthium* essential oil. After 7 days, the animals were sacrificed and the isolated small intestines were incubated to obtain adult forms of parasites and to count them. The best nematocidal effect was observed at a dose of 500 mg/kg of body weight, at which the reduction in the number of larvae was 66.07%. A positive control with albendazole showed a 65.97% reduction in the number of larvae [19].

Another study tested the potential of an ethanolic extract from *A. absinthium* against the eggs of *Ascaris suum* (porcine roundworm, parasite of pigs) and against the larvae of *Trichostrongylus colubriformis* (parasite of rabbits). The extract showed significant lethal activity against both types of parasites [77].

Another parasite that has attracted the attention of scientists is *Haemonchus contortus*. It is a nematode whose hosts are ruminants. Most often, it infests sheep and goats, in which it causes anaemia, which adversely affects the health of these farm animals [104]. In the search for new substances capable of controlling infestations with this parasite, research has been carried out in many centres around the world using extracts from *A. absinthium*.

One study evaluated the effectiveness of an aqueous and an ethanolic extract from *A. absinthium* against *H. contortus*. The study involved a parasite motility inhibition test (in vitro), which demonstrated the direct effect of the extracts on adult nematode individuals, and a test for reducing the number of parasite eggs in host faeces (in vivo). In the motility test, both extracts showed a significant anthelmintic effect, with the ethanolic extract being more effective than the aqueous one. In an in vivo study carried out on sheep, the ethanolic extract on day 15 of the experiment showed a decrease in the number of faecal eggs by 90.46% for a dose of 2 g/kg BW and 82.85% for a dose of 1 g/kg BW. The aqueous extract proved to be less active also in this test. The maximum reduction in egg count of 80.49% was observed at a dose of 2 g/kg BW. The likely reason for the greater activity of the ethanolic extract from *A. absinthium* may be the better solubility in alcohol of the compounds responsible for the anthelmintic effect [75].

Another study has also shown that aqueous and ethanolic extracts from *A. absinthium* (in an in vitro experiment) reduce the motility of *H. contortus*. An extract given to sheep for 8–10 days caused complete expulsion of the parasite without visible toxicity in the animals [78].

In contrast to the positive results of the above experiments, one cannot overlook the study conducted in 2011 in which an ethanolic extract from *A. absinthium* proved to be ineffective against the same parasite. At a dose of 1000 mg/kg BW, no beneficial effect was observed in another animal model (gerbils) [105]. Due to the conflicting results of the studies, further research is required on the impact of *A. absinthium* extracts on *H. contortus*.

### 7.2. New Possible Applications Substantiated by Scientific Research

#### 7.2.1. Antiprotozoal Effect

Extracts from A. absinthium showed antiprotozoal activity against numerous pathogens: Plasmodium berghei, P. falciparum, Naegleria fowleri, Trypanosoma brucei, T. cruzi, Leishmania aethiopica, L. donovani, L. infantum, Trichomonas vaginalis, and Entamoeba histolytica.

One experiment (in 1990) examined the antimalarial effect of *A. absinthium* on an animal model. Erythrocytes (1 × 10^7^) infected with *Plasmodium berghei*, the malaria-causing protozoan, were intravenously administered to mice. After seven days, the rodents were given leaf extracts of *A. absinthium*. The aqueous extract was given only orally, while the ethanolic extract was given orally, subcutaneously, or intraperitoneally. All the extracts produced a reduction in parasitemia, but the best inhibition effect of 96.2% was obtained with the ethanolic extract at a dose of 74 mg/kg. The results of the study showed that preparations with *A. absinthium* could be an effective agent in the treatment of malaria [79].

A further study conducted in 2010 confirmed also the antimalarial action of *A. absinthium*. The in vitro study performed on chloroquine-resistant (K1) and chloroquine-sensitive (CY27) strains of *Plasmodium berghei* grown with human erythrocytes showed that a hydro-ethanolic extract of *A. absinthium* herb exhibited significant antimalarial activity. IC_50_ values were determined after the plant extracts had been administered by the lactate dehydrogenase method. They were 0.46 μg/mL for the K1 strain and 0.195 μg/mL for the CY27 strain.

In a study in vivo, in turn, female BALB/c mice were injected with 1 × 10^7^ erythrocytes infected with *Plasmodium berghei*. On four consecutive days, the rodents were intravenously given 100 or 200 mg/kg *A. absinthium* herb extract. With the dose of 200 mg/kg, there was an 83.28% reduction in the numbers of protozoa in the blood of the mice. The authors of the study indicated that the essential oil may have been responsible for the antimalarial activity [80].

A different study tested the activity of *A. absinthium* against several pathogens that cause Chagas disease, malaria, and leishmaniasis: *Trypanosoma brucei* and *T. cruzi, Leishmania infantum* and *Plasmodium falciparum*, respectively. Various concentrations of *A. absinthium* herb extract were tested against these protozoa. The extract showed low activity only against *T. brucei* [82].

Another protozoon against which the activity of extracts of the plant was checked is *Naegleria fowleri*. It causes protozoal meningitis and encephalitis. The study checked the activity of an ethanolic and an aqueous extract of *A. absinthium*, containing only the sesquiterpenoid lactone fraction. After applying the diluted extracts onto a suspension of *N. fowleri*, even a 100% inhibition of protozoal growth was observed, the effect being clearly dependent on the concentration [84].

In the next stage of the research, it was found that the aqueous extract of the herb of *A. absinthium* showed a lethal effect on *Plasmodium falciparum*. Using a 35-fold diluted aqueous extract (1 g/mL), an 89.8% inhibition of protozoal growth was achieved [85].

Another study looked at the use of *A. absinthium* in leishmaniasis. The essential oil was isolated from the plant and tested for its ability to control the protozoa—*Leishmania aethiopica* and *L. donovani*. The results of the study showed an activity that inhibited the development of these microorganisms. The minimal inhibitory concentration (MIC) for both microorganisms in the promastigote form was 0.1565 μL/mL, while for amphotericin B the value of MIC was 0.0244 μL/mL. The EC_50_ (half maximal effective concentration) value determined for the amastigote form was 42 nL/mL for *L. donovani* and 7.94 nL/mL for *L. aethiopica*, with amphotericin as the control, for which the EC_50_ was 0.018 μL/mL and 0.047 μL/mL, respectively. The results of the study may indicate that the compounds contained in *A. absinthium* can be potential, effective agents for the treatment of leishmaniasis [83].

As part of subsequent, newer studies (2011), the use of *A. absinthium* in the treatment of leishmaniasis caused by *Leishmania infantum* infection was also examined and, additionally, the effectiveness of plant extract in the treatment of Chagas disease caused by *Tripanosoma cruzi* was also investigated. The ED_50_ value and the percentage of inhibition of protozoan growth after addition of *A. absinthium* extracts were tested. The tested extracts had a strong antiprotozoal effect. To find the compounds responsible for this activity profile, the flavonoids—casticin and artemetin, and the sesquiterpenoid lactone—hydroxypelenolide, were tested; unfortunately, none of the tested compounds showed significant lethal effects [34].

These tests were continued by testing various concentrations of *A. absinthium* essential oil. High mortality rates were found for both protozoa, reaching 96% for *L. infantum* and 99% for *T. cruzi* [24].

Some other experiments focused on *Trypanosoma cruzi* and *Trichomonas vaginalis* (vaginal trichomonad that causes trichomoniasis). The essential oil obtained from *A. absinthium* was divided into nine fractions. All of the tested fractions showed lethal activity against both protozoa. After examining the quantitative and qualitative compositions of the fractions showing the highest activity (fraction VLC1 and VLC2), it was found that (*E*)-caryophyllene and 3,6-dihydrochamazulene may be responsible for the antiprotozoal profile due to their high amounts in these fractions.

Further research into the antiprotozoal activity of *A. absinthium* is supported by the lack of cytotoxicity towards healthy human cells, which has been confirmed in an in vitro study on the HS5 bone marrow stromal cell line [26].

Clinical studies have also been performed on the efficacy of *A. absinthium* in amoebiasis caused by *Entamoeba histolytica* (amoeba that causes dysentery). Patients (numbering 25) with intestinal amoebiasis were given a 500 mg capsule containing powdered *A. absinthium* herb three times a day for 15 weeks. The remedy brought relief to the patients at various stages of the disease, and complete eradication was achieved in 70% of them [81].

#### 7.2.2. Antimicrobial and Antifungal Activities

*A. absinthium* exhibits effective antifungal and antibacterial activities. Numerous studies have shown the sensitivity of various microorganisms to the compounds contained in *A. absinthium*. Among them are bacteria such as: *Arthrobacter* spp., *Bacillus cereus*, *B. mycoides*, *B. subtilis*, *Clostridium perfringens*, *Enterobacter aerogenes*, *Enterococcus faecalis*, *Escherichia coli*, *Haemophilus influenzae*, *Klebsiella oxytoca*, *K. pneumoniae*, *Listeria monocytogenes*, *Micrococcus lylae*, *Proteus mirabilis*, *Pseudomonas aeruginosa*, *Shigella sonnei*, *Staphylococcus aureus* and the fungi: *Aspergillus niger*, *Candida albicans*, *Fusarium culmorum*, *F. graminearum*, *F. moniliforme*, *F. oxysporum fs. lycopersici*, *F. sambucinum*, *F. solani*, *Rhizoctonia solani*, *Saccharomyces cerevisiae* var. *chevalieri*, *Sclerotinia* sp. Extracts from the plant are particularly effective against Gram-positive bacteria. Gram-negative bacteria have greater resistance, which is associated with the presence of an outer phospholipid membrane that serves as an additional protective barrier of these microorganisms [16].

One experiment tested the standard strains of Gram-negative bacteria: *Escherichia coli*, *Pseudomonas aeruginosa*, *Klebsiella pneumoniae*, *Shigella sonnei*, and Gram-positive bacteria: *S. aureus*, *Clostridium perfringens*, *L. monocytogenes*, and also bacterial strains isolated from patients’ stools or wounds (*E. coli*, *E. aerogenes*, *P. aeruginosa*, *K. oxytoca*, *P. mirabilis* and *S. aureus*). In the experiment, after adding *A. absinthium* essential oil to bacterial suspensions, the MIC and MBC (minimal bactericidal concentration) were determined. The range of MIC values ranged from <0.08 mg/mL for *P. mirabilis* and *E. aerogenes* isolated from stool and for *P. aeruginosa* and *S. aureus* isolated from wounds, up to 2.43 mg/mL for *K. oxytoca* isolated from stool. The MBC of essential oil ranged from 0.08 mg/mL against *E. aerogenes* isolated from stool and *S. aureus* and *K. oxytoca* isolated from wounds, to 38.8 mg/mL against *L. monocytogenes*. At a dose of approximately 40 mg/mL, the essential oil from *A. absinthium* showed to be effective against all the bacteria tested in vitro. To be administered in vivo, this dose would probably have to be much higher [20].

*A. absinthium* is a species that can probably be used also in patients with postoperative wound infections caused by *Staphylococcus aureus*. This is indicated by the results of tests carried out on rats that were cut on their back and the wound was infected with *S. aureus*. The wound was treated topically with a hydro-alcoholic extract of *A. absinthium* herb. The number of bacteria in the test group was 3 × 10^5^ cfu/wound, while in the control group 7 × 10^6^ cfu/wound. The extract showed significant bactericidal effectiveness, which the authors attribute to the compounds found in the essential oil of the plant [86].

A subsequent study tested the effectiveness of a hydro-alcoholic extract from the herb of A. absinthium against Pseudomonas aeruginosa, Haemophilus influenzae, Bacillus subtilis, B. cereus, Klebsiella pneumoniae and Staphylococcus aureus. The experiment used the diffusion-disc method and the parameter determined was the diameter of the growth inhibition zone. The A. absinthium extract showed an activity against all the pathogens except K. pneumoniae. With a 750 mg/mL dose of the extract, the inhibition zone was 11.9 mm for Pseudomonas aeruginosa, 18.4 mm for Haemophilus influenzae, 14.4 mm for Bacillus subtilis, 20.4 mm for B. cereus, and 15.9 mm for Staphylococcus aureus [87].

There has also been a study on the effect of *A. absinthium* on the fungi *Fusarium moniliforme, F. oxysporum fs. lycopersici* and *F. solani*. Antifungal activity was determined by the diluted agar method with the addition of 0.05 mg/mL methyltetrazolium salt. The percentage inhibition of colony growth was adopted as the measured value. The tests involved checking the activity of essential oil derived from cultivated *A. absinthium* plants, from wild plants in natural habitats, and from commercial oils. The best activity was shown by a commercial oil and one of the cultivated populations. Seven of the fifteen oils tested were active against *F. oxysporum*, six oils against *F. solani*, and only two against *F. moniliforme*, below ED_50_ = 1 μg/mL. Authors claimed that the strongest effects were found for thujone free cultivated *A. absinthium* plants [24].

As part of other experiments, the antimicrobial activity of oil from *A. absinthium* herb was tested against *Enterococcus hirae, Escherichia coli, Staphylococcus aureus*, and against *Candida albicans* and *Saccharomyces cerevisiae* var. *chevalieri*. The sensitivity of the pathogens was assessed by liquid diffusion. The extract was antifungal towards both yeast species. The study did not prove antibacterial activity against the bacteria used [18].

Still other experiments using the diffusion-disc method examined the activity of *A. absinthium* essential oils against the bacteria: *Staphylococcus aureus* (strains sensitive and resistant to methicillin) and *Listeria monocytogenes*, and the fungi: *Fusarium graminearum, F. culmorum, F. oxysporum, Sclerotinia* sp. and *Rhizoctonia solani*. The essential oils tested in the experiment had been extracted from plants harvested in various parts of Tunisia; however, the results showed that habitat location had no influence on the effects produced by the oils. The strongest antibacterial activity was obtained against a methicillin-sensitive strain of *S. aureus*, for which the diameter of growth inhibition was as large as 25 mm (inhibition zone for the positive control, tetracycline, was also 25 mm). For the methicillin-resistant *S. aureus*, the largest inhibition zone was 16 mm, while for *L. monocytogenes* it was a maximum of 20 mm (positive control—24 mm). In terms of antifungal activity, the oil was observed to have a significant effect on *F. graminearum, F. culmorum* and *F. oxysporum*. The essential oil extracted from a plant collected in the area of Jerissa (city located in the north-eastern part of Tunisia) proved to be additionally active against *Sclerotinia* sp. and *Rhizoctonia solani*. The authors of the study attributed the antibacterial and antifungal activities to the main component of the oil—chamazulene, but indicated that further research was needed [22].

In recent years, the attention of scientists has been directed towards the search for new strategies to overcome antibiotic resistance of individual strains of bacteria. One of the methods of increasing the sensitivity to antibiotics is the simultaneous use of substances that inhibit the activity of membrane efflux pumps (EPI—Efflux Pump Inhibitor). Such compounds prevent the removal of the drug from inside the bacteria and increase the effectiveness of therapy [89]. In 2011, international studies were conducted to determine the antimicrobial potential of *A. absinthium*, with particular emphasis on the effect of caffeoylquinic acid derivatives on the ability to inhibit pump activity in Gram-positive bacteria. Pathogens such as *S. aureus*, *E. faecalis*, *E. coli* and *C. albicans* were tested. The results revealed at least two active compounds isolated from *A. absinthium*—chlorogenic acid, whose antimicrobial activity was low, and 4,5-di-O-caffeoylquinic acid, which caused inhibition of pump activity in Gram-positive bacteria. The obtained data indicate that *A. absinthium* may be a source of substances capable of reducing the antibiotic resistance of pathogens [37].

Similar studies have demonstrated the activity of *A. absinthium* essential oil against the bacteria: *Arthrobacter* spp.*, Bacillus mycoides*, *Micrococcus lylae*, *Pseudomonas aeruginosa* and against many species of fungi. The studies provided evidence for the greatest inhibition of pump activity against: *Alternaria alternata*, *Fusarium oxysporum*, *F. sambucinum*, *F. solani* and *Aspergillus niger* [88].

#### 7.2.3. Anti-Ulcer Effect

One of the new directions of *A. absinthium* activity covered by research has been the anti-ulcer activity, studied in laboratory animals. A group of rats were induced to develop ulcers using acetylsalicylic acid. Extracts from the whole herb and roots of *A. absinthium* obtained using various solvents (carbon tetrachloride, chloroform, methanol, ethanol, hexane) were administered to the rodents both before and after they had received acetylsalicylic acid. The extracts from the plant did not affect the activity of mucin; however, they caused a significant reduction in the volume of gastric juice, a decrease in the secretion of gastric acid and pepsin, and a decrease in digestion rate [89].

#### 7.2.4. Hepatoprotective Effect

Another new direction of *A. absinthium* activity studied has been the hepatoprotective activity. Experiments were conducted to investigate the protective effect of a hydro-methanolic extract from *A. absinthium* herb on hepatocytes in rats. Liver damage in the animals was caused by the administration of acetaminophen (paracetamol) and carbon tetrachloride. After prophylactic two-day administration of the plant extract at a dose of 500 mg/kg BW, reduced levels of asparagine and alanine aminotransferases were found in the serum, in comparison with the control group. There was also a 20% reduction in mortality due to acetaminophen administration. In the next stage of the study, *A. absinthium* extracts were tested to verify if they could be used to reduce liver damage after administration of acetaminophen and carbon tetrachloride. Acetaminophen-induced hepatotoxicity was significantly reduced after three doses of extract at 500 mg/kg, but administering them did not affect carbon tetrachloride-induced damage. Attempts were thus made to explain the hepatoprotective mechanism of action of *A. absinthium*. To this end, the effect of the plant extract on the duration of sleep induced by pentobarbital in mice and mortality induced by strychnine were investigated. Evidence was found of sleep prolongation and increased mortality, suggesting inhibition of the activity of liver microsomal enzymes [16,90].

In 2016, another experiment confirming the hepatoprotective effect of *A. absinthium* was conducted. Twenty male rats, after being divided into four equal groups, received by gavage, respectively, a saline solution, or 10, 50 or 100 mg/kg/day powdered methanolic extract from *A. absinthium* herb. Samples of rodent blood were then taken and liver indicator levels compared with control samples (before using the extract and saline solution). The best results were obtained for a dose of 50 mg/kg, at which a significant decrease in the levels of alanine aminotransferase and asparagine aminotransferase was found. Taking advantage of the capacity for reducing iron (II) ions, the antioxidant properties were also measured, which increased significantly in rats receiving the extract at 50 mg/kg. The study thus proved the hepatoprotective properties of wormwood extracts [91].

Evidence confirming the protective effect of *A. absinthium* on liver cells has been provided by further experiments conducted on mice. In these laboratory animals, liver damage was induced by administering carbon tetrachloride, or immunologically by injecting lipopolysaccharide. The rodents were given an aqueous extract of *A. absinthium*. The results were very promising: significant reduction in the levels of liver enzymes, inhibition of lipid peroxidation, and restoration of the activity of superoxide dismutase (SOD) and glutathione peroxidase (GPx) in both chemically and immunologically induced liver damage. In addition, a significant reduction in the number of pro-inflammatory mediators—TNF*α* and IL-1, was observed in the immunological model. Histopathological and other liver tests also showed a reduction in the number of inflammatory cells [36].

#### 7.2.5. Anti-Inflammatory Effect

Another new direction of *A. absinthium* activities under investigation was the anti-inflammatory activity. In one study, a methanolic extract from the herb of the plant was given orally to mice at doses of 300 mg/kg, 500 mg/kg, and 1000 mg/kg. A positive control group received 300 mg/kg acetylsalicylic acid, while a negative control group received a 0.9% sodium chloride solution. All of the mice were then intravenously given carrageenan, which induced an inflammatory response. Anti-inflammatory activity was estimated volumetrically by measuring rodent paw volume using a plethysmometer. There was a 41% reduction in oedema volume, but it was short-lived, less intense and delayed compared to the effect of acetylsalicylic acid. The authors of this work do not exclude that *A. absinthium* has a significant anti-inflammatory effect; however, further studies are needed [92].

A similar study was conducted in 2014. During the similar experiment, mice were prophylactically given *A. absinthium* essential oil at 2, 4, or 8 mg/kg, or an aqueous extract from the plant at 50, 100, or 200 mg/kg; acetylsalicylic acid served as a positive control. Inflammation of rodent paws, as in the previous study, was induced with carrageenan. The results showed that for the oil at 4 and 8 mg/kg there was a significant reduction in paw oedema, which indicated anti-inflammatory activity of *A. absinthium* [25].

Researchers from four research centres were successful in isolating from *A. absinthium* the flavonoid—5,6,3′,5′-tetramethoxy-7,4-hydroxyflavone (p7F), which was tested in in vitro and in vivo models for anti-inflammatory effects. The compound was tested for its effects on the production of nitric oxide (NO), prostaglandin E2 (PGE2), tumour necrosis factor (TNF-*α*), as well as for the expression of inducible nitric oxide synthase (iNOS), cyclooxygenase-2 (COX-2), and its effect on collagen-induced arthritis. It was found that p7F inhibited the expression of iNOS and COX-2, and also reduced the production of PGE2 and NO in lipopolysaccharide-stimulated cells of the RAW 264.7 line (cell line of monocytes and macrophages). After administering p7F to the mice in which inflammation had been induced by collagen, there was a decrease in the level of TNF-*α* and inhibition of the NF-κB pathway. The compound also prevented intracellular accumulation of reactive oxygen species. The results suggest that p7F isolated from *A. absinthium* may find use in the treatment of inflammatory diseases [42].

In 2010, a randomized clinical trial was conducted that examined the effect of *A. absinthium* extract on the plasma level of tumour necrosis factor (TNF-*α*), which is a marker of ongoing inflammation in the course of Crohn’s disease (ChL-C). Patients with ChL-C who were included in the study were divided into two groups of ten. The control group continued standard therapy and received a placebo. The study group, in addition to standard therapy, took orally 250 mg of dried, powdered *A. absinthium* herb extract three times a day for six weeks. The absinthin content in a single dose was 0.32–0.38%. The patients had their plasma TNF-*α* levels measured just before starting the treatment, then after three weeks and after six weeks. In the study group, there was a decrease in TNF-*α* concentration from the initial value of 24.5 pg/mL to 8.0 pg/mL after six weeks. In the control group, the value of 25.7 pg/mL (week 0) dropped to 21.1 pg/mL (week 6). The change in Crohn’s Disease Activity Index (CDAI) was also assessed and it was found to have decreased from 275 to 175 in the group of patients treated with *A. absinthium*, and from 282 to 230 in the group receiving the placebo. Based on the conducted research, it can be assumed that *A. absinthium* may be an effective plant in the treatment of inflammatory diseases in which elevated TNF-*α* concentration is observed [38].

Cardamonin—a chalcone isolated from *A. absinthium*, has been tested for its potential anti-inflammatory activity. To this end, the effect of cardamonin on lipopolysaccharide-induced release of nitrites and expression of iNOS and COX-2 proteins was assessed on two cell lines: THP-1 (monocyte cell line of acute monocytic leukaemia) and RAW 264.7 (cell line of mouse macrophages). Using the western blot technique, the compound was also tested for its effect on phosphorylation of mitogen-activated protein kinases (MAP): the ERK, JNK, and p38 MAP kinases, and on the activation of the NFĸB pathway. The results showed inhibition of NO release and inhibition of iNOS expression, depending on cardamonin concentration, which led to the suppression of inflammation. The compound had no effect on the expression of COX-2, phosphorylation of MAP kinases, or phosphorylation of NFĸB. There was, however, inhibition of the NFĸB pathway by direct inhibition of DNA transcription factor. Induction of IFN-γ activated iNOS was also inhibited [39].

Another study examined whether *A. absinthium* would reduce inflammation induced with carrageenan and the venom of *Montivipera xanthina* (Middle Eastern viper species) in rats. Half an hour before the administration of venom or carrageenan, the rodents were intraperitoneally given *A. absinthium* herb extract. The results indicated that administration of the plant extract at 25 and 50 mg/kg significantly inhibited venom-induced paw oedema, while doses of 12.5, 25 and 50 mg/kg were effective in carrageenan-induced inflammation [93].

Anti-inflammatory activity can also be induced by caruifolin D contained in *A. absinthium*, which has been proven in studies conducted in Beijing. This compound, by inhibiting the inflammatory process, influenced the protection of nerve structures. The exact mechanism of action is described in the sub-section “Neuroprotective effect” [41].

#### 7.2.6. Immunomodulatory Effect

Extracts from *Artemisia absinthium* have also been the subject of research on their immunomodulatory activity. One experiment examined whether *A. absinthium* extract had an effect on the maturation of dendritic cells in laboratory mice. Dendritic cells were treated with an ethanolic extract of *A. absinthium* herb for 18 h and then examined in a flow cytometer. The extract was found to have a positive effect on dendritic cell maturation by increasing the level of surface expression of CD40 protein, which acted as a marker co-stimulating dendritic cells and the induction of cytokines. The study also assessed the degree of proliferation of allogeneic T-lymphocytes using a mixed lymphocyte reaction (MLR) and enzyme-linked immunoassay (ELISA). During the MLR, allogeneic T-lymphocytes were mixed with dendritic cells isolated from the mice treated with the *A. absinthium* herb extract. It was found that at 100 μg/mL of extract the proliferation of T-lymphocytes was reduced by 78.2% relative to the control. A significant increase in IL-10 levels was also observed. The obtained results may serve to justify the traditional use of *A. absinthium* in immune disorders [94].

Another study proved that polysaccharides isolated from *A. absinthium* herb showed immunostimulatory activity by inducing TH1 response and stimulating nitric oxide production by mouse peritoneal macrophages [50]. Th1 helper lymphocytes, which support the body’s cellular response, participate in TH1 response. Bactericidal macrophages and Tc-lymphocytes are activated, and IgG class antibodies are produced, which activate the complement system. IL-2 interleukins and interferon-γ are also produced [106]. The immune response elicited by polysaccharides isolated from the herb of *A. absinthium* is particularly effective against intracellular viruses and bacteria [107].

#### 7.2.7. Cytotoxic Effect

*A. absinthium* extracts have also been tested for anti-tumour activity. A methanolic extract from the herb of the plant was used to treat breast cancer cells of the MDA-MB231 line (non-oestrogen-responsive line) and MCF-7 breast adenocarcinoma cells (oestrogen-responsive line). After three days of exposure, a 50% inhibition of MDA-MB231 cell proliferation at 20 g/mL and a 50% inhibition of MCF-7 cells at 25 g/mL were demonstrated. Based on the results obtained, it was concluded that *A. absinthium* might be a potential source of new compounds limiting the development of breast cancer and breast adenocarcinoma [95].

The cytotoxicity of *A. absinthium* essential oil has also been tested. In vitro studies were performed on six cell lines: A548 (lung adenocarcinoma cell line), NCI-H292 (non-small-cell lung cancer cell line), HCT116 (colon cancer cell line), MCF-7 (breast adenocarcinoma cell line), SK-MEL-5 (melanoma cell line), and HS5 (bone marrow stromal cell line—control). After multiplying the cells in the medium, their reaction to *A. absinthium* essential oil and its individual fractions was examined. The tested lines were found to be sensitive to the essential oil and/or its individual fractions in a dose-dependent manner, with the best effect observed against the SK-MEL-5 and HCT116 lines, and the weakest against the MCF-7 line. The authors of the study associate the occurrence of the cytotoxic effect mainly with the presence of (*E*)-caryophyllene and/or germacrene D because of the high concentration of these compounds in the oil [26].

#### 7.2.8. Analgesic Effect

As part of other experiments, in parallel with the testing of anti-inflammatory activity, as described above, *A. absinthium* was also evaluated for its analgesic effect. The following scheme of the experiment was used: the methanolic extract of the herb of the plant was administered orally to the mice in the study group, at doses of 300 mg/kg, 500 mg/kg or 1000 mg/kg. A positive control group received 300 mg/kg acetylsalicylic acid, while a negative control group received a 0.9% sodium chloride solution. The analgesic effect was assessed based on the delay in withdrawing the tail from water heated to 51 °C. A rapid onset of analgesic action was seen at all doses, but it was less pronounced compared to that of acetylsalicylic acid. As in the study of anti-inflammatory activity, it was found that *A. absinthium* could produce analgesic effects, but more experiments were required [92].

The analgesic properties of extracts from *A. absinthium* have been confirmed in the writhing test and the hot plate test with mice in other studies. Before conducting the two tests, rodents were given *A. absinthium* essential oil at doses of 2, 4, or 8 mg/kg, or an aqueous extract from the leaves of the plant at doses of 50, 100, or 200 mg/kg. There was a significant reduction in writhing episodes in the writhing test and a longer delay in pain response in the hot plate test compared to the control with morphine, which may confirm the analgesic profile of *A. absinthium* [25].

#### 7.2.9. Neuroprotective Effect

The neuroprotective potential of *A. absinthium* has also been assessed. To this end, an experiment was designed in which obstruction of the middle cerebral artery in rats was caused with a nylon thread. After 90 min, the thread was removed and the normal blood perfusion restored for 24 h. The temporary closure of the middle cerebral artery led to infarction, lipid peroxidation, reduction in glutathione levels, and a decrease in the activity of catalase and superoxide dismutase. There was a lack of locomotor coordination and short-term memory impairment among the tested rats. An attempt was made to reduce the losses by prophylactic oral administration of 100 mg/kg and 200 mg/kg methanolic extract from the herb of *A. absinthium*. The results proved to be promising—there was evidence of a reduction in oxidative stress, in the extent of brain damage, and in behavioural disorders, which indicates that extracts from *A. absinthium* herb may be a potential agent in the prevention of stroke [96].

The results of another study indicate that the neuroprotective effect of *A. absinthium* may be associated with the anti-inflammatory activity of the sesquiterpenoid dimer—caruifolin D, present in the plant. In the experiment, this compound significantly inhibited the production of neuro-inflammatory mediators in BV2 microglial cells. By inhibiting the production of reactive oxygen species, caruifolin D also showed antioxidant activity; a decrease in protein kinase C and stress-activated kinases (JNK) was also demonstrated. The authors of the study proved that caruifolin D in *A. absinthium* had neuroprotective and anti-inflammatory potential. The researchers proposed this compound as potential for use in the treatment of inflammatory neurological diseases such as Alzheimer’s or Parkinson’s [41].

Further studies assessed the role of *A. absinthium* aqueous extract in the neuroprotection of the brain during prolonged exposure to lead. Among the group of rats that had been exposed to lead, the number of dopaminergic neurons in the *substantia nigra* decreased by 50%, and the number of astrocytes in the frontal cortex increased by 48%. It was proved that a four-week treatment of rodents with an aqueous extract of *A. absinthium* at a dose of 200 mg/L reversed most of the lesions occurring in glia and in the dopaminergic system [97].

One of the latest studies (2016) focused on the neuroprotective potential of *A. absinthium* herb extracts in the oxidative stress caused by mercury. Damage in rat brains was induced by oral administration of mercury (II) chloride (HgCl_2_) at a dose of 5 mg/kg. The rodents were then given 500 mg/L aqueous *A. absinthium* extract. A significant decrease in malondialdehyde (MDA) was recorded—by 26.99% in the cerebellum, by 31.81% in the cerebral cortex, and by 80.70% in the striatum; induction of catalase activity in the cerebellum was also evident. The plant extract also restored the activity of antioxidant enzymes—superoxide dismutase, glutathione peroxidase, glutathione reductase and thioredoxin HgCl_2_ reductase [108].

#### 7.2.10. Antidepressant Effect

*A. absinthium* has also been tested as a plant with potential antidepressant activity. For this purpose, the forced swim test and the tail suspension test were carried out on mice. In the first test, 125, 250, 500, or 1000 mg/kg methanolic extract of *A. absinthium* was injected intraperitoneally into the mice from the study group. A positive control group received 5 or 10 mg/kg imipramine solution, and a negative control group a 0.9% saline solution. The rodents were placed in a cylinder filled with water and the duration of immobility was measured during the last four minutes of the experiment. The *A. absinthium* extract significantly shortened the period of animal’s immobility compared to the negative control group. The effect observed with an extract dose of 1000 mg/kg BW was comparable to the effect after application of 5 mg/kg imipramine. The second test consisted in hanging the mouse by its tail. After initial vigorous movements, periods of immobility during a five-minute observation were recorded. The results of the second test showed that the *A. absinthium* extract significantly and in a dose-dependent manner reduced the time of immobility of the mice, and that the effect caused by the 500 mg/kg dose of the extract represented the same antidepressant activity as that of imipramine [98].

#### 7.2.11. Procognitive Activity

A study was conducted to check whether the historically known use of *A. absinthium* in the treatment of memory disorders and reduced concentration was justified. Because it was known that cholinergic receptors were involved in cognitive functions, the study examined the activity of the binding of compounds extracted from *A. absinthium* to muscarinic and nicotinic receptors. Ethanolic extracts from two batches of harvested plants were applied to the prepared homogenate of human cortical brain cells; the ability to displace [3H]-(N)-nicotine and [3H]-(N)-scopolamine from the receptors was determined. The results of the study showed that *A. absinthium* had significant affinity for both muscarinic and nicotinic receptors, meaning that preparations from the plant can show procognitive effects [99].

#### 7.2.12. Neurotrophic Action

A study on the effects of *A. absinthium* herb extract on the nerve growth factor (NGF) in PC12D cells (cell line of rat pheochromocytoma tumour) has also been conducted. Three types of extract from the plant—methanolic, ethanolic, and aqueous, were tested. All of them showed neurotrophic activity, increasing neurite development through NGF induction. The strongest effect was proven for the methanolic extract, followed by the ethanolic one, while the aqueous extract produced a weak effect. The obtained results indicate that extracts from *A. absinthium* herb may gain significance in the treatment of neurodegenerative disorders [100].

#### 7.2.13. Cell Membrane Stabilizing Effect

The protective effect of *A. absinthium* herb extracts on the haemolysis of human erythrocytes has also been studied. Absorbance of a red blood cell solution with the addition of sodium chloride was observed in the absence or presence of a crude aqueous extract of the *A. absinthium* herb. The extract from the plant proved to be a good stabilizer of erythrocyte membranes, which can protect cells from hypotonic shock [101].

#### 7.2.14. Antioxidant Effect

The herb of *A. absinthium* contains numerous flavonoids and other phenolic compounds that can potentially determine its antioxidant activity.

In one experiment, the antioxidant activity of *A. absinthium* was tested using the DPPH method. The plant material was extracted by successively using solvents of different polarity (70% methanol, petroleum ether, chloroform, ethyl acetate, n-butanol). It was demonstrated that the antiradical activity depended on the type of solvent and the concentration of extracts. The best results were obtained using the acetate extract, then the extracts produced with methanol, n-butanol, chloroform, and ether. The authors of the study concluded that the concentration of phenols and flavonoids influenced the antioxidant activity of *A. absinthium* [102].

As part of another experiment, several tests were performed to test the activity of a methanolic extract from *A. absinthium* herb. In the DPPH test, the IC_50_ value for radical scavenging activity was 612 μg/mL. The IC_50_ values for ascorbic acid, quercetin and butylated hydroxyanisole (BHA) used as controls were 1.26 μg/mL, 1.32 μg/mL and 13.49 μg/mL, respectively. The methanolic extract from *A. absinthium* also showed a reduction potential in the reaction with iron (III) ions, where the effect was proven to depend on the concentration of the extract. Evidence was also found for a good capacity to chelate iron(II) cations. The capacity for scavenging nitric oxide (NO) and hydrogen peroxide (H_2_O_2_) was also tested. The IC_50_ value for nitric oxide was 1.77 mg/mL and for quercetin (as a positive control) the IC_50_ value was 17.01 μg/mL. The *A. absinthium* extract also showed the ability to scavenge H_2_O_2_, where the IC_50_ value was 24.3 μg/mL, while for control samples with ascorbic acid and BHA these values were IC_50_ = 21.4 μg/mL and IC_50_ = 52.0 μg/mL, respectively. Flavonoids (12.4 mg quercetin equivalent/g extract) and phenolic compounds (194.9 mg gallic acid equivalent/g extract) were isolated from the tested extract. The tests prove that the significant antioxidant activity of *A. absinthium* is determined by the considerable concentrations of flavonoids and phenolic compounds [98].

A further study showed that extracts of *A. absinthium* herb from Spanish crops had a stronger antioxidant effect in the DPPH test than the individually tested flavonoids—artemetin and casticin, and the sesquiterpenoid lactone—hydroxypelenolide. In addition, the antioxidant effect of hydroxypelenolide was stronger than that of the flavonoids tested. The results of the work suggest that the antioxidant activity of *A. absinthium* is governed by the synergy of the compounds present in the plant [34].

Antioxidative properties of methanolic extracts of *A. absinthium* herb were also tested using the DPPH method with plants harvested from various regions of Tunisia. The results indicated that the effectiveness of the antioxidant activity of the extracts was related to the location of the habitat from which the plants were harvested. The strongest effect was found for extracts from the herb of plants collected in the northern part of Tunisia (IC_50_ = 9.38 mg/mL). The second test method tested the reduction potential for iron (III). Herb extracts were able to reduce iron(III) ions regardless of where the plants had been harvested; moreover, the EC_50_ values obtained were lower than for the ascorbic acid control [22].

In another study using the DPPH and ABTS (2,2′-azino-bis(3-ethylbenzothiazoline-6-sulfonic acid) methods, significant antioxidant properties of *A. absinthium* essential oil were found [20].

A subsequent study examined in vitro the activity of scavenging superoxide anions, hydrogen peroxide, hydroxy radicals and nitric oxide, as well as the reduction potential of *A. absinthium* herb extracts. In addition, an in vivo study was performed in which, after giving a methanolic extract of the plant to mice, bilateral carotid artery occlusion was performed, reperfusion was restored, and then the superoxide dismutase activity, concentration of thiobarbituric acid reactive substances (TBARS) and glutathione content were determined by colorimetric methods. The results of the tests in vitro showed significant antioxidant activity in relation to all the compounds tested. In vivo, significant inhibition of oxidative stress was found in the central nervous system after oral administration of 100 or 200 mg/kg of *A. absinthium* herb extract. The amount of TBARS also decreased and the concentrations of superoxide and glutathione dismutases increased, which indicates the possibility of using extracts of the plant as an antioxidant [103].

In yet another study, the DPPH scavenging test was carried out on extracts from in vitro suspension cultures of *A. absinthium* [34,109]. The highest antioxidant activity (82.7%) and maximum accumulation of flavonoids (1.89 mg quercetin equivalent/g DW) and phenols (3.57 mg gallic acid equivalent/g DW) were demonstrated for a 30-day suspension culture [35].

### 7.3. Importance in Veterinary Pharmacology

The herb of *A. absinthium* is characterized by a strong taste that can change the sensory experience among animals consuming fodder. Adding the dried plant to the feed for ruminants has been found to stimulate their appetite [73,109].

One of the experiments examined whether *A. absinthium* had a stimulating effect on appetite in sheep. A flock of sheep (16 animals) was divided into four groups; in each of them the animals had a different diet. The animals in the first group received 300 g/kg dry matter (DM) basic concentrates and 700 g/kg DM rice straw. In the second group, 50 g/kg DM straw was replaced with silage containing *A. absinthium*; in the third group this was 100 g/kg DM, and in the fourth group 150 g/kg DM silage. The results of the study showed that enriching the diet of sheep with *A. absinthium*-containing silage significantly increased the amount of feed consumed. Improved digestion, increased nitrogen retention, and an increase in the number of microorganisms involved in the assimilation of nitrogen were also observed [73].

The effect of *A. absinthium*-containing silage on feed intake was also tested among Hanwoo steers. The same research method was used and improvements in nutrient supply and digestion were observed, as well as faster animal growth and improvements in carcass quality and the amount of fatty acids [74].

## 8. Applications in Cosmetology

In addition to the undeniable therapeutic properties, *A. absinthium* has also found application in cosmetics used for scalp, face, and hair care.

CosIng (Cosmetic Ingredient database)—a European database gathering data on cosmetic ingredients, allows the use of *Artemisia absinthium* in five forms. Among them there are skin care products, fragrances, and substances with antibacterial activity (Table 4) [110].

Raw materials obtained from the plant are used as components of cosmetic products such as shampoos, face serums, masks, essences, tonics, moisturizing creams with an SPF filter, and under-eye patches. These forms of cosmetics are used to protect, cleanse, and moisturize the skin, as well as to remove skin imperfections. They are produced mainly with extracts of the herb of the plant or distilled oil; also included is the filtrate obtained after fermentation of the leaves by *Lactobacillus* sp.

Products containing *A. absinthium* can be found in the offers of foreign companies worldwide. South Korean, Russian and American cosmetics producers are leaders among them.

## 9. Applications in the Food Industry

*A. absinthium* is the main ingredient in absinthe, which is a high-proof alcoholic beverage that was particularly popular in the 19th and 20th centuries because of its psychoactive properties due to the high *α*- and *β*-thujone content [15,111]. Information on the safety and mechanism of action of both compounds is presented later in this review.

Dried aerial parts of the plant are used to produce absinthe. In a traditional recipe, wormwood, along with other herbs, is macerated. The obtained macerate of a greenish colour with a slightly stinging, strongly bitter taste is subjected to distillation, leading to a reduction in the amount of bitter compounds. In the last stage, the distillate is diluted with water and the product has a characteristic light green colour [15].

The European Food Safety Authority (EFSA) states that in the European Union, the *α*- and *β*-thujone content in alcoholic beverages, including absinthe, must not exceed 10 mg/kg for spirits with an ethanol content higher than 25%, or it must not exceed 35 mg/kg in bitter spirits [112].

Wormwood is also added to wines to give them aroma and bitterness. Vermouths are a popular type of wine containing *A. absinthium*. The whole herb together with *A. absinthium* roots is used for their production; also used are other aromatic or bitter herbs (e.g., *Salvia officinalis*, *Coriandrum sativum*, *Citrus aurantium* var. *amara*) and spices (e.g., *Syzygium aromaticum*, *Cinnamomum zeylanicum*, *Zingiber officinale*). In the United States, vermouths are used to make cocktails, while in Europe they are served without any admixtures [113].

In small quantities, *A. absinthium* is recommended for seasoning meat, vegetable soups and fresh vegetables. It is also used as a dye and flavouring in the traditional Korean rice cake *“green songpyeon”*, which is an integral part of the celebration of the *“chuseok”* thanksgiving festival. In Morocco, *A. absinthium* is added to mint tea [31].

## 10. Safety of Use

It is worth pointing out the dangers of drinking absinthe. Its consumption initially causes the feeling of well-being and hallucinations, slowly leading to a depressive stage. Chronic abuse of the spirit has been described as absinthism, characterized by blindness, tremors, hallucinations, and significant deterioration of the mental state. The degeneration that is observed in the advanced stage causes convulsions and can even lead to death.

Currently, however, the influence of the substances present in *A. absinthium* herb on the development of absinthism is being questioned. The most probable hypothesis is that absinthism is misdiagnosed alcoholism because all the symptoms characteristic of absinthism can be attributed to ethanol itself [114].

*A. absinthium* is a species rich in compounds that show toxic effects. These are *α*- and *β*-thujone, with *α*-thujone being thought to be two to three times more harmful. The likely mechanism of action of these compounds is interaction with the GABA receptor of chloride channels [15]. The compounds exhibit neurotoxic activity leading to hyperactivity, tremors and tonic convulsions [16]. These symptoms have been confirmed in studies on laboratory animals (mice). Intraperitoneal injection of *α*-thujone into rodents induced tremors. Convulsions did not occur if diazepam or phenobarbital was administered prophylactically [115].

EFSA emphasizes, however, that the *α*- and *β*-thujone content in the essential oil of the plant ranges from 0% to 70.6%, which will also result in the occurrence or absence of side effects [31].

Wormwood should not be used if the patient has gastric or duodenal ulcers, biliary obstruction, liver disease, or if he is allergic to plants of the family Asteraceae. This species should not be used during pregnancy and breastfeeding because, as shown in experiments on pregnant rats, *Absinthii herba* hinders embryo implantation and reduces the number of births. An overdose of preparations containing the plant may result in vomiting, diarrhoea and urinary retention [11,30].

The potential effect of skin irritation and potential acute toxicity of the essential oil of the plant were investigated in 2014. After testing in healthy volunteers, researchers found no skin irritation after applications of undiluted *A. absinthium* oil. Acute toxicity tests in a group of mice did not show increased animal mortality after oral application of the essential oil; however, neurological, muscular and gastrointestinal problems were observed [20].

As potentially dangerous compounds, EFSA lists *α*- and *β*-thujone, absinthin, and anabsinthin. The summarizing conclusions regarding *A. absinthium* contain the information that the plant can be safely used as a basic substance. It has a known toxicological profile, and the compounds that were previously considered harmful are currently being investigated as medicinal substances [31].

The FDA (U.S. Food and Drug Administration) lists *A. absinthium* as an allergenic species. The source of allergens is the pollen, which can also be present in extracts of the plant [116].

## 11. Biotechnological Research

Although there are no problems with obtaining *A. absinthium* from natural habitats or by cultivation, attempts are being made at finding biotechnological solutions. They can undoubtedly bypass the problems associated with the chemical variability of the raw materials derived from the plant—the herb and the essential oil. Biotechnological research to date has been concerned with endogenous production of secondary metabolites and the development of micropropagation protocols.

In 2013, one of Pakistani research groups proposed a method of producing *A. absinthium* secondary metabolites by cultures in vitro. In the first stage aimed at establishing the cultures, they tested the Murashige-Skoog (MS) medium enriched only with the addition of 0.5–5.0 mg/L thidiazuron (TDZ), or a combination of TDZ with 1.0 mg/L naphthalene-1-acetic acid (NAA), or with 1.0 mg/L indole-3-acetic acid (IAA). Thus prepared substrates were used to initiate cultures from leaf explants. Callus obtained on MS media supplemented with 1.0 mg/L TDZ and 1.0 mg/L NAA was passaged for biomass growth. Further experiments on the growth and production of secondary metabolites involved the use of suspension cultures established from 35-day callus cultures. Biomass was grown in Erlenmeyer flasks containing MS medium with 1.0 mg/L TDZ and 1.0 mg/L NAA. The biomass was taken for analysis at 3-day intervals during 42-day cultivation cycles. Phytochemical analysis was performed by HPLC. Seedlings from germinating seeds constituted the control sample. The amount of phenolic acids in the suspension was 3.57 mg/g (control: 2.75 mg/g), and the amount of flavonoid compounds was 1.77 mg/g (control: 1.20 mg/g). The results of the study showed that *A. absinthium* suspension cultures might be a good potential source of phenolic acids and flavonoids [35].

Scientists from the same research group also studied the effect of light and its absence on the accumulation of secondary metabolites in *A. absinthium* suspension cultures. The cultures were grown on MS medium supplemented with 1 mg/L TDZ and 1 mg/L NAA. The dynamics of biomass growth were measured during 39-day cultivation cycles. Cultures grown in the presence of continuous artificial light with a radiation intensity of 40 mol·m^−2^·s^−1^ reached their maximum, 3.9-fold, increase in dry biomass on day 30, whereas cultures grown in the dark, a 3.7-fold increase, on day 27 of cultivation. In addition, in vitro cultures grown in the presence of light were found, using UV-VIS spectroscopy methods with the addition of Folin-Ciocalteu reagent, to have elevated levels of phenols, while those with the addition of aluminium chloride—elevated levels of flavonoids. The results of the study showed that appropriate lighting conditions for *A. absinthium* cultivation had a positive effect on the efficiency of secondary metabolite production [117].

A Hindu research group has developed a protocol for micropropagation of *A. absinthium* by indirect organogenesis. The most effective for callus growth was MS medium with 0.5 mg/L 2,4-dichlorophenoxyacetic acid (2,4-D) and 0.5 mg/L kinetin (Kin). Maximal induction of shoots from callus was found on a medium containing 4.5 mg/L BAP (6-benzylaminopurine) and 0.5 mg/L NAA. With a different quantitative combination of growth regulators, i.e., 1.5 mg/L BAP and 0.5 mg/L NAA, a pronounced induction of many shoots from nodal explants was observed. The most beneficial medium promoting rhizogenesis was one with the addition of 0.5 mg/L indole-3-butyric acid (IBA) [118].

## 12. Conclusions

*A. absinthium* is a species with a very important position in the history of Asian and European medicine; described in medieval Europe as *“the most important master against all exhaustions”*, it was mainly used to treat digestive tract diseases and worm infestations.

Nowadays, this species has the status of a pharmacopoeial species in the European allopathic as well as homeopathic therapies. Currently the species occupies an important place in the traditional European and Asian medicine.

In modern times, *A. absinthium* has been the object of numerous studies on the chemistry of raw materials derived from it—the herb and the essential oil, as well as numerous studies on the biological activity of extracts.

Research on the chemistry of the plant has identified a large number of compounds in the herb, including most of all the presence of essential oil with a very rich but variable chemical composition, bitter sesquiterpenoid lactones, flavonoids, other bitterness-imparting compounds, azulenes, phenolic acids, tannins and lignans.

Research on biologically active extracts from the herb and/or individual isolated compounds and/or essential oil has drawn attention to the mechanism of action of these raw materials in known classical applications. It has also provided evidence for numerous, very valuable, previously unknown, new directions of biological activity of the raw materials—antiprotozoal, antibacterial, antifungal, anti-ulcer, hepatoprotective, anti-inflammatory, immunomodulatory, cytotoxic, analgesic, neuroprotective, antidepressant, procognitive, neurotrophic, cell membrane stabilizing, and antioxidant effects.

Both phytochemical and pharmacological tests are carried out by research centres located all over the world.

The species is also used with great success as a source of cosmetic raw materials, in Southeast Asia, North America (USA) and Europe, in particular. The long-known significance of the species in the food industry, as a base in the production of alcoholic beverages (absinthe and vermouth wines) and as a valuable spice, is not decreasing. The species has also become the subject of biotechnological research on the production of bioactive compounds and the possibility of micropropagation using established in vitro cultures.

The proven new directions of the biological activity of extracts from the herb and of individual isolated compounds and the essential oil of wormwood substantiate the medieval claim that *Artemisia absinthium* is: *“the most important master against all exhaustions”*.

## Figures and Tables

**Figure 1 plants-09-01063-f001:**
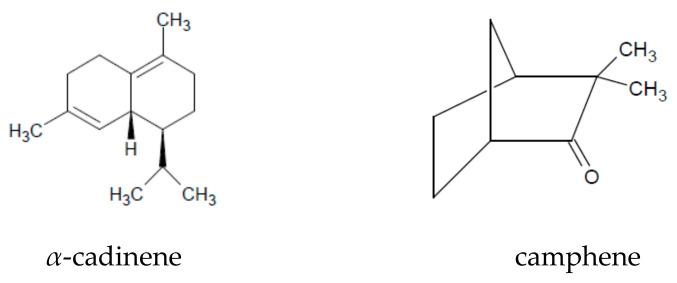
Chemical structure of volatile compounds characteristic for *A. absinthium* herbal essential oil.

**Figure 2 plants-09-01063-f002:**
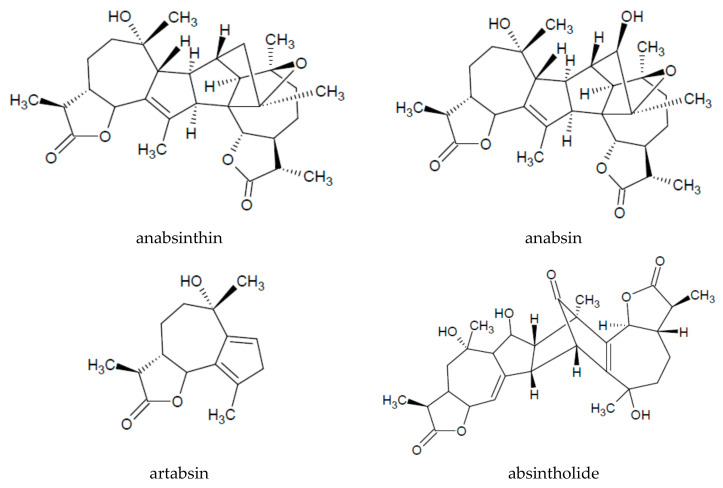
Chemical structure of sesquiterpenoid lactones characteristic for *A. absinthium.*

**Table 1 plants-09-01063-t001:** The chemical composition of *A. absinthium* essential oil.

Chemical Groups/Compounds	References
Monoterpenoids	
(*E*)-6,7-epoxyocimene, (*Z*)-6,7-epoxyocimene, (*Z*)-carveol, carvacrol, geranyl pentanoate, geranial, *p*-menth-3-en-9-ol, neryl acetate	[18]
(*E*)-epoxyocimene	[19]
(*Z*)-epoxyocimene	[16,19]
1,8-cineole	[1,16,17,18,19,20,21,22,23,24,25]
geranyl 2-methylbutanoate, neryl 2-methylpropanoate, linalyl 3-methylbutanoate, geranyl 3-methylbutanoate, bornyl 3-methylbutanoate, linalyl butanoate, (*Z*)*-β*-epoxyocimene, fenchone, (*E*)-sabinene hydrate, isobornyl acetate, isobornyl propanoate, pulegone, *α*-fenchene	[23]
neryl 2-methylbutanoate	[21,23]
2-*β*-pinene, lyratyl acetate	[25]
linalyl 3-methylbutanoate	[18,23]
neryl 3-methylbutanoate	[21,23]
terpinene-4-ol, (*E*)-sabinene hydrate, (*E*)-sabinol	[22,23]
thujyl alcohol	[1,21]
*allo-*ocimene	[18,20]
*Artemisia* ketone, 3-methylbutanoate, (*E*)-thujone, phellandrene, isothujyl acetate, pinene, (*E*)-verbenol, (*Z*)-thujone	[21]
borneol, (*Z*)-nerolidol, (*Z*)-verbenol, (*E*)-*β*-ocimene, (*Z*)-sabinene hydrate, *α*-terpinyl acetate, *p*-cymen-8-ol, terpinolene, *α*-terpinene	[22]
chrysanthenol	[19,24]
(*Z*)-chrysanthenol	[15,26]
(*Z*)-expoxyocimene	[15,21,24,26]
phellandrene epoxide, thujol	[18,24]
eugenol	[18,22,23,27]
geraniol	[1,18,22,27]
iso-3-thujanol	[27]
geranyl isovalerate, lavandulyl acetate, *allo*-ocimene, *β*-linalool	[20]
camphene	[1,19,22,23]
camphor	[16,22,23,24,26,28]
carvone	[21,22]
lavandulol	[20,22,23,24]
limonene	[22,23,24,25]
linalool	[18,19,21,22,23,24,26]
myrcene	[16,21]
neral, geranyl acetate, neryl acetate, (*Z*)*-β*-ocimene	[18,22]
nerol	[16,18,21,22,27]
bornyl acetate	[15,22,23,24]
chrysanthenyl acetate	[15,16,21,24,26,27]
(*Z*)-chrysanthenyl acetate	[16,18]
linalyl acetate	[16,22,23,24]
sabinyl acetate	[15,18,20,22,24]
(*E*)-sabinyl acetate	[16,21,23,28]
thujyl acetate	[1]
*p*-cymene	[1,16,18,22,23]
linalyl propionate	[23,24]
sabinene	[18,20,21,22,23,24]
thymol	[24]
santolinatriene	[23,25]
(*Z*)-linalooloxide	[22,23,24]
(*E*)-linalool oxide	[22,24]
epoxyocymene	[21,24]
tricyclene	[29]
*α*-phellandrene	[16,18,20,23]
*α*-pinene	[1,16,18,19,22,23,24,25]
*α*-terpineol	[1,18,20,24,25]
*α*-thujene	[18,22,23]
*α*-thujone	[1,15,16,18,23,24,30]
*β*-phellandrene	[1,18,22]
*β*-myrcene	[18,19,20,23,28]
*β*-pinene	[16,21,22,23,28]
*β*-thujone	[1,15,18,20,23,24,27,28,30,31]
γ-terpinene	[18,20,22,23]
**Sesquiterpenoids**	
(*E*)-nerolidol, ar-curcumene, diepi-*α*-cedrene, bisabolol oxide, *α*-copaene, *β*-gurjunene	[18]
(*E*,*E*)-farnesyl acetate, (*E*,*E*)-farnesal, (*Z*,*E*)-*α*-farnesene, (*E*,*E*)-farnesyl 3-methylbutanoate, 7-*α*-silphiperfol-5-ene, *allo*-aromadendrene, bicyclogermacrene, (*Z*)*-α*-bisabolene, cyperene, epi-*β*-santalene, hexahydrofarnesyl acetone, petasitene, pethybrene, presilphiperfol-7-ene, (*E*)-nerolidyl propanoate, silfinen-1-en, silphiperfol-6-ene, humulene oxide II, *α*-cedrene, *α*-gurjunene, *α*-isocomene, *α*-santalene, *α*-(*E*)-bergamotene, *β*-bisabolene, *β*-eudesmol, *β*-isocomene, *β*-santalene, γ-humulene	[23]
elemol, guaiazulene, cadinene, *α*-himachalene	[1]
germacrene D	[16,19,22,26]
caryophyllene	[1,24]
curcumene	[21]
nerolidol, (*E*)*-β*-farnezene	[25]
spathulenol	[18,27]
bisabololoxide B	[27]
caryophyllene oxide	[1,21,22,23,24,25,27]
(*E*)-caryophyllene	[26]
*α*-bisabolene, *α*-calacorene, γ-curcumene, γ-muurolene	[22]
*α*-bisabolol	[18,22,23,27]
*α*-humulene	[18,22,23]
*α*-copaen	[22,23]
*β*-bourbonene	[18,23]
*β*-elemene	[19,23]
*β*-caryophyllene	[18,19,20,23]
*β*-selinene	[18,22,23,24,26]
γ-gurjunene	[18,23]
γ-cadinene	[18,22]
δ-cadinene	[18,23,25]
**Diterpenoids**	
1-(*E*)-8-isopropyl-1,5-dimethyl-nona-4,8-dienyl-4-methyl-2,3-dioxa-bicyclo(2, 2, 2)oct-5-ene, iso-1-(*E*)-8-isopropyl-1,5-dimethyl-nona-4,8-dienyl-4-methyl- 2,3-dioxa-bicyclo(2, 2, 2)oct-5-ene	[1,32]
vulgarol A, vulgarol B	[18]
**Phenylpropanoids**	
methyleugenol	[27]
estragole	[27]

**Table 2 plants-09-01063-t002:** The chemical composition of *A. absinthium* herb.

Chemical Group	Compound	References
**Sesquiterpenoid lactones**	absintholide	[9,16]
	absinthin	[1,9,15,16,21,31]
	anabsin, ketopepenolid-A, *β*-santonin	[16]
	anabsinthin	[16,21,31]
	arabsin, ketopelenolide, santonin related lactones	[21]
	artabin	[16,21]
	artabsin	[15,16,21]
	artenolide, deacetyloglobicin, isoabsinthin, parishine B and C	[9]
	germacranolide, hydroxypelenolide	[34]
	caruifolin D	[41]
	matricin	[9,16]
**Bitter principles**	24-zeta-ethylcholesta-7,22-dien-3-*β*-ol, artamaridin, artamaridinin, artamarin, artamarinin, quebrachitol	[21]
**Azulenes**	3,6-dihydrochamazulene	[26]
	7-ethyl-1,4-dimethylazulene	[19]
	7-ethyl-5,6-dihydro-1,4-dimethylazulene	[16]
	azulene	[1,21]
	chamazulene	[18,21,22,23,26,28]
	dihydrochamazulene isomer	[16]
	prochamazulenogen	[21]
**Flavonoids**	quercetin-3-rutinoside	[36]
	5,6,32′,5′-tetramethoxy 7,4′-hydroxyflavone	[21,42]
	5-hydroxy-3,3′,4′,6,7-pentamethoxyflavone, glycosides of quercetin	[21]
	apigenin, quercetin dihydrate, flavone, kaempferol, catechin, myristin, naryngenin	[22]
	artemetin	[1,21,34]
	*Artemisia* bis-isoflavonyl dirhamnoside, *Artemisia* isoflavonyl glucosyl diester	[1]
	casticin	[34]
	quercetin	[16]
	rutoside	[16,21]
**Chalcones**	cardamonin	[38,39]
**Coumarins**	herniarin	[27]
	coumarin	[22]
**Phenolic acids**	1′,3′-O-dicaffeoylquinic acid, 1′,5′-O-dicaffeoylquinic acid, 3′,5′-O-dicaffeoylquinic acid, 4′,5′-O-dicaffeoylquinic acid, 5′-O-caffeoylquinic acid	[37]
	chlorogenic acid	[16,21,36,37]
	ferulic acid	[22,31]
	gallic acid	[22,35]
	caffeic acid	[16,21,22,31,35]
	coumaric acid, salicylic acid	[16]
	*p*-coumaric acid, rosmarinic acid, tannic acid	[22]
	syringic acid, vanillic acid	[16,22]
**Organic acids**	succinic acid, malic acid, (E)-cinnamic acid	[22,31]
**Fatty acids**	9- hydroxy-(*E*)-10,12-octadecadienoic acid, 13- hydroxy-(*E*)*,* (*E*)-9, 11-octadcadienoic acid, epoxyoleic acid, linoleic acid, oleic acid, palmitic acid, stearic acid	[1]
	dodecanoic acid	[18]
**Sterols**	3,11-dimethyldodecan-1,7-dioic acid-1-*β*-D-glucopyranosyl-6′- octadec-9′′-enoate, lanost-24-en-3*β*-ol-11-one-28-oic acid-21,23 *α*-olide-3*β*-D-glucopyranosyl-2′-dihydrocaffeoate-6′- decanoate	[40]
**Fatty acid glycosides**	ethyl linoleate, methyl linoleate, ethyl palmitate, methyl palmitate	[23]
**Tannins**	nd *	[16,21,22,31]
**Lignans**	nd	[16,21]
**Carotenoids**	nd	[16,21]
**Resinous substances**	nd	[31]
**Polysaccharides**	nd	[43]
**Other compounds**	(*5Z*)-2,6-dimethylocta-5,7-diene-2,3-diol	[19]
	(*Z*)-2,6-dimethylocta-5,7-diene-2,3-diol	[24,26]
	(*Z*)-jasmone, 2-ethyl-4-methyl-1,3-pentadienylbenzene, 3-octanol, bicyclo[2.2.1]-hept-2-en-7-ol, (*E*)-3-hexenyl butyrate, (*Z*)-3-hexenyl butyrate, benzeneacetaldehyde, fraganol, 3,7-dimethyl-2-metyl propanoic acid	[18]
	1H-benzocycloheptene, 4-hexen-1-ol, benzenemethanol, benzene, 1-butanol, en-in-dicycloether, (*E*)-photonerol,	[25]
	(*E*)-nuciferyl 2-methylpropanoate, albene, *(E)*-nuciferyl butanoate, hexanal, (*Z*)-nuciferyl propanoate	[23]
	trimethoxybezoic acid	[1]
	(*E*)-3-hexenyl butyrate	[19,26]
	nuciferol butanoate, nuciferol propionate	[21]
	silica	[31]
	stigmast-5,22-dien-3*β*-ol-21-oic acid-3*β*-glucopyranosyl-2′- octadec-9′′-enoate, tricosan-14-on-1,4-olide-5-eicos-9′-enoate	[40]

* nd—no data.

**Table 3 plants-09-01063-t003:** Pharmacological properties of *A. absinthium.*

Activity	Mechanism of Action	References
**Stimulating digestion**	Change in postprandial haemodynamics in the gastric digestive phase with increased hyperaemia, probably due to the effects of bitter compounds contained in the herb of the plant.	[72]
**Stimulating appetite**	Enrichment of sheep fodder with silage containing *A. absinthium* increases the amount of fodder consumed, improves digestion, induces nitrogen retention and has a positive effect on the development of microorganisms involved in nitrogen assimilation.	[73]
	Improvement in nutrient supply and digestion, faster growth, improvement in carcass quality and amount of fatty acids among Hanwoo steers.	[74]
**Anthelmintic**	Extracts from *A. absinthium* cause paralysis and/or death of *Haemonchus contortus* nematodes and reduce the number of the parasite’s eggs in the host’s faeces.	[75]
	Lethal effect on *Trichinella spiralis* larvae.	[19,76]
	Lethal effect of *A. absinthium* ethanolic extract on *Ascaris suum* eggs and *Trichostrongylus colubriformis* larvae.	[77]
	Lethal effect on *Haemonchus contortus* tested in vivo; reduction in its mobility in vitro.	[78]
**Antiprotozoal**	Lethal effect of aqueous and ethanolic extracts from *A. absinthium* on *Plasmodium berghei*.	[79]
	Lethal effect of the essential oil on *Plasmodium berghei.*	[80]
	Lethal effect of A. absinthium on Entamoeba histolytica.	[81]
	Some lethal activity against *Trypanosoma brucei*.	[82]
	Lethal activity against the promastigota and amastigota forms of the protozoa *Leishmania aethiopica* and *Leishmania donovani*.	[83]
	Lethal activity in vitro against *Leishmania infantum* and *Trypanosoma cruzi*	[24,34]
	Lethal effect of the essential oil on *Trypanosoma cruzi* and on *Trichomonas vaginalis*. The compounds likely to be responsible for this activity are (E)-caryophyllene and 3,6-dihydrochamazulene.	[26]
	Inhibition of *Naegleria fowleri* growth by sesquiterpenoid lactones in *A. absinthium.*	[84]
	Lethal effect of *A. absinthium* aqueous extract against *Plasmodium falciparum*.	[85]
**Antibacterial** **Antifungal**	Growth inhibition by the essential oil from A. absinthium and its lethal activity against: Escherichia coli, Pseudomonas aeruginosa, Klebsiella pneumoniae, Staphylococcus sonnei, Staphylococcus aureus, Clostridium perfringens, Listeria monocytogenes, Enterobacter aerogenes, Klebsiella oxytoca, and Proteus mirabilis.	[20]
	Bactericidal activity of *A. absinthium* essential oil components against *Staphylococcus aureus*.	[86]
	Lethal effect of A. absinthium extract on Pseudomonas aeruginosa, Haemophilus influenzae, Bacillus subtilis, Bacillus cereus, and Staphylococcus aureus.	[87]
	Inhibition of growth of *Fusarium oxysporum*, *Fusarium solani* and *Fusarium moniliforme* by the components of *A. absinthium* essential oil.	[24]
	Inhibition of growth of Saccharomyces cerevisiae var. chevalieri and Candida albicans.	[18]
	Inhibition of growth of the bacteria *Listeria monocytogenes* and methicillin sensitive/resistant *Staphylococcus aureus*, and the fungi *Fusarium graminearum*, *Fusarium culmorum*, *Fusarium oxysporum*, *Sclerotinia* sp. and *Rhizoctonia solani* by chamazulene in the essential oil.	[22]
	Some bactericidal activity of chlorogenic acid and efflux pump inhibition (EPI) by 4,5-di-O-caffeoylquinic acid isolated from *A. absinthium*.	[37]
	Lethal action against the fungi Alternaria alternata, Fusarium oxysporum, Fusarium sambucinum, Fusarium solani and Aspergillus niger, and the bacteria Arthrobacter spp., Bacillus mycoides, Micrococcus lylae, Pseudomonas aeruginosa.	[88]
**Anti-ulcer**	Decrease in gastric juice volume, reduction in gastric acid and pepsin secretion, and decrease in digestion rate.	[89]
**Hepatoprotective**	*A. absinthium* extracts inhibit liver microsomal enzymes that are responsible for the metabolism of xenobiotics.	[90]
	Methanolic extracts from the herb of the plant protect liver cells by reducing ALAT and ASPAT levels, and by reducing oxidative damage.	[91]
	Protection of the liver due to the immunomodulatory and/or antioxidant properties of *A. absinthium*.	[36]
**Anti-inflammatory**	Reduction of inflammatory oedema in mice after administration of the essential oil or methanolic extract from *A. absinthium*.	[25,92]
	Inhibition of the expression of nitric oxide synthase and cyclooxygenase-2, reduction in the production of prostaglandin E2, nitric oxide and tumour necrosis factor (TNF-α), reduction in the accumulation of reactive oxygen species by 5,6,3′,5′-tetramethoxy-7,4-hydroxyflavone isolated from *A. absinthium*.	[42]
	Suppression of tumour necrosis factor (TNF-α) by compounds present in *A. absinthium*. Among the compounds likely to be responsible for the anti-inflammatory activity of the plant are the chalcone cardamonin, flavonoids, artemisinin, and semisynthetic artesunate.	[38]
	Cardamonin isolated from *A. absinthium* inhibits the NFĸB pathway by direct inhibition of DNA transcription factors, which leads to reduced NO release.	[39]
	Reduction of paw oedema in rats given carrageenan and venom of *Montivipera xanthina* after application of *A. absinthium* extract.	[93]
**Immuno-stimulating**	Induction of dendritic cell maturation by increasing the level of CD40 surface expression and by induction of cytokines.	[94]
	Induction of TH1 immune response and stimulation of nitric oxide production by macrophages.	[43]
**Cytotoxic**	Inhibition of proliferation of breast cancer cells of MDA-MB-231 and MCF-7 lines.	[95]
	The essential oil, in particular (E)-caryophyllene and/or germacrene D, is toxic to tumour lines A548, NCI-H292, HCT116, MCF-7, SK-MEL-5.	[37]
**Analgesic**	Reduction of temperature-induced pain in mice.	[92]
	Reduction in episodes in the writhing test and delay in pain response in the hot plate test in mice after administration of *A. absinthium* essential oil or aqueous extract.	[25]
**Neuroprotective**	Methanolic extract from *A. absinthium*, because of its antioxidant potential, reduces brain damage, inhibits of lipid peroxidation, and restores the activity of enzymes involved in reducing oxidative stress. Flavonoids and phenolic acids in the plant are probably responsible.	[96]
	Protective effect of *A. absinthium* aqueous extract on glial cells and the dopaminergic system when exposed to lead.	[97]
	Caruifolin D in Absinthii herba inhibits the production of pro-inflammatory microglia mediators and reactive oxygen species, and also inhibits protein C kinase and stress-activated kinases.	[41]
**Antidepressant**	Shortening of the period of mouse immobility in the forced swim test and in the tail suspension test.	[98]
**Procognitive**	Affinity for human muscarinic and nicotinic receptors responsible for cognitive functions.	[99]
**Neurotrophic**	Methanolic, ethanolic and aqueous extracts from *A. absinthium* induce the nerve growth factor (NGF), which stimulates development of neurites.	[100]
**Stabilizing cell membranes**	Hydro-alcoholic extract from *A. absinthium* prevents haemolysis of erythrocytes.	[101]
**Antioxidant**	Antioxidant activity of flavonoids and phenolic compounds in *A. absinthium*.	[35]
	Reducing properties of polyphenols towards free radicals.	[102]
	*A. absinthium* contains active compounds that allow electron donation, which prevents oxidation of structures by reactive oxygen species.	[98]
	Synergistic antioxidant effects of the compounds present in the plant.	[34]
	Methanolic extracts from *A. absinthium* herb have a significant reduction potential.	[22]
	*A. absinthium* essential oil has the ability to scavenge radicals in DPPH and ABTS tests.	[20]
	Reducing properties of *A. absinthium* extract and the ability to capture superoxide and hydrogen peroxide anions, hydroxy and nitric oxide radicals; inhibiting oxidative stress, reducing the concentration of TBARS, increasing the concentration of superoxide and glutathione dismutases.	[103]

**Table 4 plants-09-01063-t004:** Use of *A. absinthium* in cosmetology as recommended by CosIng database.

CosIng Data	Description	Functions
*Artemisia absinthium* extract	extract from the whole wormwood herb	skin conditioning
*Artemisia absinthium* herb extract	extract from the blooming herb of wormwood	perfuming
*Artemisia absinthium* oil	volatile oil obtained from the whole wormwood plant	antimicrobial
*Artemisia absinthium* herb oil	essential oil obtained from the blooming wormwood herb	perfuming
*Lactobacillus*/*Artemisia absinthium* leaf extract ferment filtrate	filtrate of the product obtained by fermentation of wormwood leaves by bacteria of the genus *Lactobacillus*	skin conditioning

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
