# Peer review of "Artemisia absinthium L.—Importance in the History of Medicine, the Latest Advances in Phytochemistry and Therapeutical, Cosmetological and Culinary Uses"

_plants, 2020, doi:10.3390/plants9091063_

Round 1
Reviewer 1 Report
Review Report
Artemisia absinthium L. (wormwood) − importance in the history of medicine, the latest advances in phytochemistry and therapeutical, cosmetological and culinary uses
This review paper on recent advances of research related to Artemisia absinthium is a well-structured and comprehensive report, summarizing available literature sources from the last two decades, including the latest publications. In addition, several relevant online databases/websites have been screened for data related to wormwood.
The structure of the manuscript follows a logical order, starting with the botanical description of the species, followed by phytochemical traits, the plant’s role in the history of medicine and in traditional medicine. Finally, probably the most significant part of the manuscript provides an overview about the position of wormwood in modern allopathy and homeopathy, reporting various pharmacological activities and mechanisms of action in a concise manner, but providing all necessary details for proper understanding.
The paper is easy to read and follow, without any over-complicated sentences. English language and style are mostly appropriate, only a few misspellings or inappropriate word choices and sentence structures should be corrected by the authors.
In summary, I think that the paper could be published after a minor revision.
Please find my detailed comments and suggestions below.
Abstract
L25-26: It would be better to use the accepted drug names Absinthii herba (Ph. Eur. 10.0) and Artemisiae absinthii aetheroleum instead of Artemisiae herba (this could refer to the source plant Artemisia vulgaris) and Artemisiae absinthium aetheroleum.
L27: … other bitterness-imparting… (instead of “another”)
L33: … scientific research, such as antiprotozoal ….
- Introduction
L50: studies have focused (“been” is not necessary here)
- General information on the species
Some items of botanical terminology should be revised here.
L79: “capillary secretory hair” - it is not clear what is meant by “capillary” here. I suggest that the authors simply write “secretory hair” or “glandular hair” or even “essential oil secreting hairs / glandular trichomes”, which are all common botanical terms, in contrast to “capillary secretory hair”
L81, 82: instead of “stalk” the use of the term “stem” would be correct
L84, 102, 228: instead of “butt-end leaves” the use of the term “basal leaves” or “base leaves” would be more appropriate
L87-89: instead of “basket” the term “capitulum” should be used, which is the usual term to describe the inflorescence type of Asteraceae species
L88: ligulate female flowers (“n” should be omitted)
L89-90: this should be rephrased probably like this: “The involucral bracts covering the capitulum are long and grey, with ensiform outer and oval inner leaves”
- Phytochemical characteristics
The authors are recommended to use the same kind of terminology throughout the manuscript, observing also some minor details as the use of parentheses and italicization of prefixes.
In Tables 1 and 2, and also in the text, the letters Z and E (indicating geometric isomers) should be italicized, and in parentheses: (Z); (E).
Although IUPAC (International Union of Pure and Applied Chemistry) still allows the use of the former cis/trans names, the use of the (Z/E)-system is becoming more and more accepted and recommended in scientific papers. Thus the consequent use of the (Z/E)-system is highly recommended for the authors throughout the manuscript (including Fig. 1).
(However, if the authors keep the prefixes cis/trans, these should be italicized, but not in parentheses.)
The names of geometric isomers can be considered complete and precise only if the position of the double bond is given with a number, too. For example, if the cis (Z) double bond is between carbon atoms 9 and 10, it should be indicated as (9Z). Authors should follow this rule, when providing the names of such compounds.
In case of aromatic substances, the abbreviations of orto-, meta- and para-positions (o-, m- and p) should also be in italics.
In case of optical isomers, the indexes R and S, indicating the absolute configuration of chiral centers should be presented like this: (R); (S). Preceding R or S, the number of the chiral center should be provided, e.g. (3R); meaning that carbon atom number 3 has R configuration as chiral center.
If a chemical compound involves both optical and geometrical isomers, the configuration indicating the optical isomer should always precede indication of the geometrical isomer; e.g. (3R,9Z)- …….
P5, Table 1, L3, Sesquiterpenoids: As follows from the above, the name (Z,E)-R-farnesene is not correct, instead the following should be written: (..R,..Z,..E)-farnesene, where the number of the appropriate carbon atom should be written instead of dots.
P6, Fig. 1: In case of (Z)-epoxyocimene, the stereochemical position of the methyl and dimethylallyl substituents is not provided. Both substituents should be depicted with dashed lines.
Some minor comments:
P 4, Table 1, L 14, Reference 35: instead of „allo-Ocimene”: allo-ocimene would be correct. (The prefixes allo-, syn- and anti- should be italicized, too.)
P 8, Table 2, Line 1: instead of „p-coumaric acis” p-coumaric acid is correct.
P8, Table 2, within „Fatty acids” the correct version of the first two names would be:
9-hydroxy-trans-trans,10,12-octadecadienic acid (not 9-hydroxyl; letter „l” is not necessary!), and
13-hydroxy-trans-trans,9,11-octadecadienic acid
- Position in modern allopathy and homeopathy
L235-236 and L244: Absinthii tinctura (not “tincturae”)
- Biological activities confirmed by scientific research
L267: “in wormwood herb” - “the” is not necessary here
L346: Erythrocytes (capital E, starting a new sentence)
7.2.1. Antiprotozoal effect
L364-367: Only this study (reference 81) reported negative results related to the antiprotozoal activity of A. absinthium extracts. This should receive stronger emphasis, and this study should be discussed contrasted to other studies (e.g. reference 83).
In addition, further details should be provided about this study, e.g. what kind of extract, prepared from which plant part etc. was used.
L365: “Trypanosoma brucei, T. cruzi, Plasmodium falciparum and Leishmania infantum, respectively” is suggested instead of the current version, in order to correspond to the order of diseases mentioned.
L380 and L425: the meaning of the abbreviation of EC50 and MBC, respectively, should be provided at first mentioning (similarly to MIC, written in full in L379).
7.2.2. Antimicrobial and antifungal activities
L426: MIC values ranged from ….
L456-460: In this study no antibacterial activity was reported. This should be contrasted to the other studies summarized here (similarly to the case mentioned in L364-367).
It would be useful to read an overall conclusion regarding the antimicrobial activity of A. absinthium. This could be the closing paragraph of this section.
7.2.4. Hepatoprotective effect
L523-524: different sentence structure is suggested: “…also measured, which increased significantly in rats receiving the extract at 50 mg/kg.”
- Applications in cosmetology
Table 4, Artemisia absinthium oil: “volatile oil obtained from the whole wormwood plant”
- Conclusions
L919-924: Authors are suggested to combine the following sentences into a single statement: “.. with a very important position in the history of Asian and European medicine;” (1st paragraph) and “The species occupies an important place in the traditional European and Asian medicine.” (2nd paragraph)
L928-930: “…chemistry of the plant has identified a large number of compounds in the herb, including the essential oil with a very rich ….flavonoids, other bitterness-imparting compounds…”

Reviewer 2 Report
The manuscript titled “Artemisia absinthium L. (wormwood) – importance in the history of medicine, the latest advances in phytochemistry and therapeutical, cosmetological and culinary uses” is a general review providing useful information about history, phytochemistry and pharmacology of Absinthe. The article is scientifically valid. It has a relative novelty, but contains useful information. The english seems good enough.
The paper is adequately presented and discussed.
Nevertheless, there is an important issue regarding the references section that need to be corrected. In particular, the references are presented in a disordered way. Their numbering is not progressive. Before Table 1, the last ref. number is [17] (line 125), while in the Table 1 the first number is [22]. So, the progressive numbering should only regard the text, without considering the tables (in facts, the first number after the table is [18], at line 136). On the contrary, before Table 2 the last number is [34] (line 154), while at the resumption of the text the number jumps to [44] (line 169), in this case counting all the references in the table.
In addition, other minor inaccuracies are present in many single references, as the presence of a semicolon at the end of lines 975, 982, 983, 1071, 1092, 1093, 1102; or the absence of the publication year (lines 1113, 1115, 1117).
In a review paper the reference section is very important. Therefore, my advice is that a careful reorganization of the entire reference section should be considered.
Author Response
REVIEWER 1
The manuscript titled “Artemisia absinthium L. (wormwood) – importance in the history of medicine, the latest advances in phytochemistry and therapeutical, cosmetological and culinary uses” is a general review providing useful information about history, phytochemistry and pharmacology of Absinthe. The article is scientifically valid. It has a relative novelty, but contains useful information. The english seems good enough.
The paper is adequately presented and discussed.
Nevertheless, there is an important issue regarding the references section that need to be corrected. In particular, the references are presented in a disordered way. Their numbering is not progressive. Before Table 1, the last ref. number is [17] (line 125), while in the Table 1 the first number is [22]. So, the progressive numbering should only regard the text, without considering the tables (in facts, the first number after the table is [18], at line 136). On the contrary, before Table 2 the last number is [34] (line 154), while at the resumption of the text the number jumps to [44] (line 169), in this case counting all the references in the table.
In addition, other minor inaccuracies are present in many single references, as the presence of a semicolon at the end of lines 975, 982, 983, 1071, 1092, 1093, 1102; or the absence of the publication year (lines 1113, 1115, 1117).
In a review paper the reference section is very important. Therefore, my advice is that a careful reorganization of the entire reference section should be considered.
Response:
Thank you for positive opinion on our work. We corrected and organized in the proper way References. References have been formatted according to the “Plants” instructions for the authors.

Reviewer 3 Report
Dear Editors,
Thank you so much for choosing me as reviewer of the manuscript plants-883653 entitled: “Artemisia absinthium L. (wormwood) – importance in the history of medicine, the latest advances in phytochemistry and therapeutical, cosmetological and culinary uses”. I hope that my comment will help Authors to improve their manuscript.
Detailed remarks concerning manuscript plants-883653 entitled: “Artemisia absinthium L. (wormwood) – importance in the history of medicine, the latest advances in phytochemistry and therapeutical, cosmetological and culinary uses”.
I suggest use in the title of the manuscript only the Latin name of the examined species.
Key words. It is not recommended to use as key words the same words or phrases used in the title of the manuscript (see the word wormwood).
The aim of the report should be clearly stated.
The short methodology (searched database, searched key words as well as the period (the searched years for literature).
References
Please go through the reference list and check your account on editorial mistakes. For example sometimes the each word of the cited manuscript tile is written with capital letter but the other times not. Please do needed changes.
Author Response
REVIEWER 2
Thank you so much for choosing me as reviewer of the manuscript plants-883653 entitled: “Artemisia absinthium L. (wormwood) – importance in the history of medicine, the latest advances in phytochemistry and therapeutical, cosmetological and culinary uses”. I hope that my comment will help Authors to improve their manuscript.
Detailed remarks concerning manuscript plants-883653 entitled: “Artemisia absinthium L. (wormwood) – importance in the history of medicine, the latest advances in phytochemistry and therapeutical, cosmetological and culinary uses”.
I suggest use in the title of the manuscript only the Latin name of the examined species.
Key words. It is not recommended to use as key words the same words or phrases used in the title of the manuscript (see the word wormwood).
Response:
Thank you. We corrected title and keywords according to your suggestion.
The aim of the report should be clearly stated. The short methodology (searched database, searched key words as well as the period (the searched years for literature).
Response:
Thank you. As you suggested, we added suitable information in the Introduction part.
References. Please go through the reference list and check your account on editorial mistakes. For example sometimes the each word of the cited manuscript tile is written with capital letter but the other times not. Please do needed changes.
Response:
Thank you. We corrected and organized in the proper way References. References have been formatted according to the “Plants” instructions for the authors.

Round 2
Reviewer 2 Report
The reference section has been properly emended. The paper, in my opinion, can be accepted in the present form.